# REAL-TIME DETECTION OF HALLUCINATED ENTITIES IN LONG-FORM GENERATION

## ABSTRACT

Large language models are now routinely used in high-stakes applications where hallucinations can cause serious harm, such as medical consultations or legal advice. Existing hallucination detection methods, however, are impractical for real-world use, as they are either limited to short factual queries or require costly external verification. We present a cheap, scalable method for real-time identification of hallucinated tokens in long-form generations, and scale it effectively to 70B parameter models. Our approach targets *entity-level hallucinations*—e.g., fabricated names, dates, citations—rather than claim-level, thereby naturally mapping to token-level labels and enabling streaming detection. We develop an annotation methodology that leverages web search to annotate model responses with grounded labels indicating which tokens correspond to fabricated entities. This dataset enables us to train effective hallucination classifiers with simple and efficient methods such as linear probes. Evaluating across four model families, our classifiers consistently outperform baselines on long-form responses, including more expensive methods such as semantic entropy (e.g., AUC 0.90 vs 0.71 for Llama-3.3-70B), and are also an improvement in short-form question-answering settings. Moreover, despite being trained only to detect hallucinated entities, our probes effectively detect incorrect answers in mathematical reasoning tasks, indicating generalization beyond entities. While our annotation methodology is expensive, we find that annotated responses from one model can be used to train effective classifiers on other models; accordingly, we publicly release our datasets to facilitate reuse. Overall, our work suggests a promising new approach for scalable, real-world hallucination detection.

## 1 INTRODUCTION

Large language models (LLMs) have seen rapid adoption in high-stakes fields such as medicine (Ayo-Ajibola et al., 2024; Henry, 2025) and law (Braff, 2025; Thomson Reuters Institute, 2025), where the reliability of model outputs is critical. A key limitation of LLMs, however, is *hallucinations*—the generation of content that is plausible-sounding but factually incorrect (Huang et al., 2025; Ji et al., 2023). In these high-stakes settings, even minor errors can have serious consequences, underscoring the need for robust hallucination detection methods.

Several recent works have addressed hallucination detection in short-form question-answering (QA) settings (Kuhn et al., 2023; Farquhar et al., 2024; Kossen et al., 2024). In these settings, completions are generally brief (1–2 sentences), contain a single atomic claim, and correctness can be unambiguously labeled. However, real-world LLM usage increasingly involves open-ended long-form generation (e.g., multi-turn medical consultations, or legal case analyses), where models produce complex, multi-paragraph responses containing numerous interconnected claims. Hallucination detection in long-form generation presents fundamentally different challenges: it no longer suffices to label entire responses as correct or incorrect; instead, systems must identify which specific segments are hallucinated in responses where correct and incorrect claims are intertwined (Yeh et al., 2025).

Existing approaches for long-form hallucination detection, such as SAFE (Wei et al., 2024b) and FActScore (Min et al., 2023), use expensive multi-step pipelines that extract atomic claims, retrieve external evidence, and verify each claim. The resulting cost and latency make these methods impractical for real-time monitoring during generation. This reveals a critical gap: the lack of

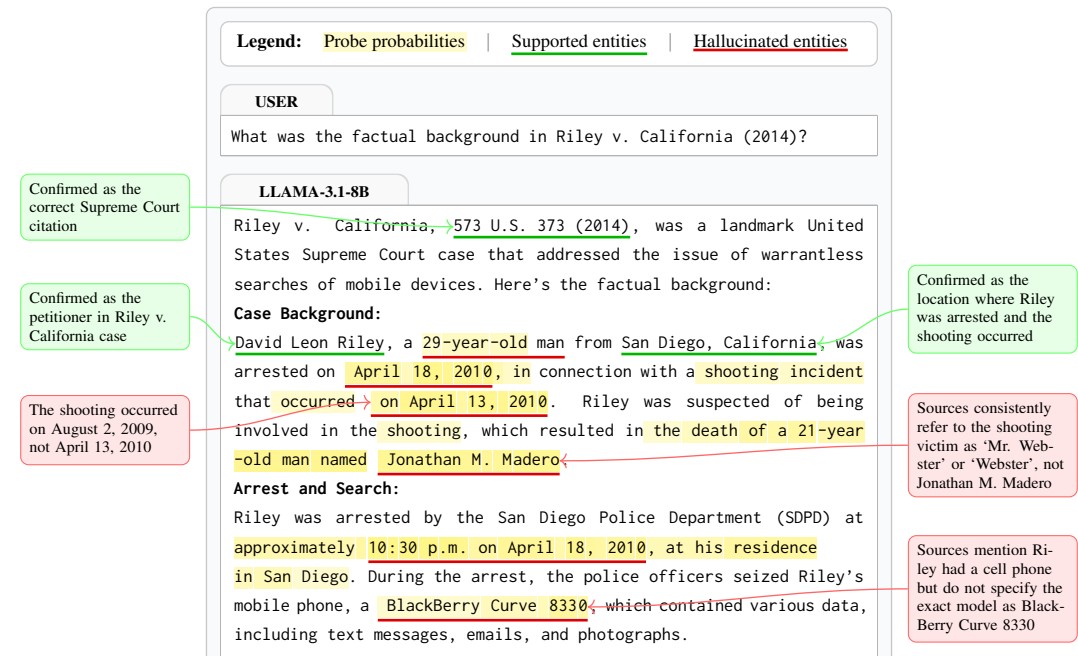

Figure 1: **An annotated example of hallucination detection in long-form legal text.** The underlines indicate entity spans labeled by our annotation pipeline: green denotes entities labeled as supported, while red denotes entities labeled as hallucinated. Hallucination detection probe scores for each token are shown as yellow highlights, with the intensity reflecting the score's magnitude (scores below 0.4 are not shown). Note that while the annotation pipeline predominantly identifies and labels *entities* (e.g., "Jonathan M. Madero", "BlackBerry Curve 8330"), this example illustrates the difficulty in cleanly separating entities from non-entities in long-form text. We notice that both our annotation pipeline and our resulting probes sometimes detect broader hallucinations, such as claims, even if they don't correspond cleanly to an entity (e.g., "at his residence in San Diego," which is indeed a fabricated detail).

*streaming classifiers* capable of flagging hallucinated content as it is produced, without requiring auxiliary verification models or external knowledge retrieval.

To address this gap, we frame hallucination detection as a token-labeling task rather than post-generation verification. Aligning with the recent work of Yeh et al. (2025), we focus on *entity-level hallucinations* (e.g., fabricated names, dates, citations) rather than claim-level. Entities have clear token boundaries and can be verified in real-time as they appear, whereas claims require post-hoc extraction that breaks token alignment and forces systems to wait for complete sentences. This design choice enables streaming detection while effectively capturing factual errors, as incorrect claims typically contain fabricated or misused entities.

The key to our approach is a data annotation technique that uses a frontier LLM augmented with web search to extract entities from model outputs and label them as factually supported or fabricated. Each token is assigned the label of its containing entity, enabling us to train lightweight linear probes that predict these labels from hidden activations. The probes run in the same forward pass and flag unsupported entities as tokens are produced with negligible computational overhead. In long-form settings, linear probes substantially outperform uncertainty-based baselines at detecting hallucinated entities, achieving 0.87 AUC on Llama-3.3-70B, compared to 0.71 AUC using a version of semantic entropy adapted to long-form generation.

We study generalization across generation settings and model families. Training probes on long-form text transfers well to short-form QA, but short-form training fails to recover long-form performance, suggesting that long-form supervision is necessary for effective monitoring. Additionally, we find that probes trained on one model can reliably detect hallucinations in other models' outputs, suggesting they capture fundamental patterns of hallucinations rather than model-specific signals.

For enhanced performance, we show that adding low-rank adapters (LoRA) during training further improves detection accuracy (0.90 AUC on Llama-3.3-70B). To maintain streaming capabilities, we employ KL regularization to balance probe performance with minimal model behavior changes. Finally, as a proof-of-concept, we demonstrate how our streaming detection approach enables real-time intervention, allowing systems to abstain from responding when hallucination risk is detected, thereby improving factual reliability.

While further work is still needed for robust practical deployment, our streaming token-level detection approach suggests a concrete path toward real-time hallucination monitoring at scale.

## 2 RELATED WORK

**Internal representation-based methods.** A growing body of work leverages models' internal states to detect hallucinations. Probing classifiers (Alain & Bengio, 2017) map intermediate model representations to target properties and have been extensively used for hallucination detection. Marks & Tegmark (2024) train linear probes to uncover truth-related directions in representation space. Recent studies (Orgad et al., 2025; Ji et al., 2024; Alnuhait et al., 2025) show that linear and MLP-based probes can predict hallucinations using hidden states before or during generation, often achieving strong AUC scores across various tasks. However, their ability to generalize to more complex settings, such as open-ended long-form generation, remains unproven. CH-Wang et al. (2024) train span-level probes to detect hallucinations on grounded tasks (e.g., document summarization), detecting inconsistencies with the provided input context (e.g., a source document).

**Uncertainty-based detection.** Hallucinations in language models can be analyzed through the lens of uncertainty estimation. The uncertainty of model predictions across an entire sequence can be quantified using the joint probability of generated tokens. To account for varying sequence lengths, previous work (Fomicheva et al., 2020; Guerreiro et al., 2023) has considered the *length-normalized generation log probability* of a model's response as an approximate measure of its uncertainty.

While many machine learning tasks involve distinct, mutually exclusive output classes (e.g., digit classification), open-ended text generation is more complex, as multiple distinct output sequences can convey essentially the same meaning. Addressing this issue, Kuhn et al. (2023); Farquhar et al. (2024) introduce *semantic entropy*: given a query, semantic entropy groups semantically equivalent answers into clusters, and quantifies how spread out the model's probability distribution is across these clusters. Kossen et al. (2024) propose a more efficient alternative: *semantic entropy probes* (SEPs)—lightweight classifiers trained to predict semantic entropy from hidden states alone, achieving competitive but lower classification performance compared to the sampling-based variant.

**External verification methods.** Methods such as SAFE (Wei et al., 2024b), FActScore (Min et al., 2023), and FacTool (Chern et al., 2023) represent a prominent approach that employs external verification for long-form hallucination detection. These methods work by first extracting claims from the generated text, then retrieving supporting evidence from external sources, and finally evaluating each claim in light of the external evidence. While effective for comprehensive verification, these pipelines incur significant computational costs and latency, making them unsuitable for real-time detection during generation. For example, a single sentence may fan out into tens of claims, each of which requires multiple search queries and LLM API calls to verify.

**Entity-based hallucination detection.** A limitation of most existing hallucination detection methods is that they operate at coarse levels of granularity, such as sentence- or passage-level classification. The recent work of Yeh et al. (2025) argues that this coarse-level assessment is inadequate for long-form generation, where accurate and fabricated information frequently coexist within a single response; for example, a generation might correctly state a historical date while fabricating the individuals involved. To enable finer-grained analysis, they propose *entity-level hallucination detection* and introduce HalluEntity, a dataset consisting of 157 biography generations from GPT 3.5, with annotations indicating which entity spans are hallucinated. We build upon this entity-level framing to develop a scalable annotation pipeline for training supervised probes.

# 3 METHODOLOGY

## 3.1 DATASET CONSTRUCTION AND ENTITY-LEVEL ANNOTATION

To train token-level hallucination detectors, we need a large, diverse dataset with precise annotations of hallucinated content within long-form outputs. This requires two steps: (1) generating diverse completions that contain both hallucinated and factual content, and (2) obtaining accurate entity-level annotations that identify which specific token spans correspond to hallucinated entities.

**Data generation.** We build upon the LongFact dataset (Wei et al., 2024b), composed of 2,280 fact-seeking prompts, and introduce LongFact++, which expands LongFact with 10 times more prompts across more diverse domains and query structures. LongFact++ incorporates four categories of prompts: topic-focused queries (e.g., queries related to the topic of "molecular mechanisms of viral DNA replication"), biographical queries about famous individuals, citation-focused prompts that encourage generation of references, and legal prompts based on landmark court cases. More details of dataset construction are provided in Appendix C.

For each target model that we study, we use the prompt sets from LongFact and LongFact++ to generate completions, creating hallucination-rich generations that serve as the source material for token-level annotation.[1]

**Entity-level annotation.** Existing verification methods like SAFE (Wei et al., 2024b) decompose generated text into atomic claims for verification, but this reformulation breaks alignment with the original token sequence needed for token-level training. Instead, we focus our annotations on *entities*—e.g., named people, organizations, locations, dates, and citations—which can be verified against external sources while preserving exact token boundaries.

For each generated completion, we use Claude 4 Sonnet with web search capabilities (Anthropic, 2025a;b) to extract and annotate specific spans within the original text. The system identifies entity spans, searches for supporting evidence, and labels each entity as "Supported," "Not Supported," or "Insufficient Information" (see the full prompt in Appendix C.5).[2] Figure 1 shows an example of a labeled completion with entity-level annotations and verification justifications.

**Label quality.** We audited label quality using several checks (details in Appendix F) and summarize the two most informative here. We recruited a human annotator to independently label a random sample of entity spans via web search; the human annotations matched the LLM's labels in 84% of cases ($n=50$). We also constructed a controlled set of hallucinations by paraphrasing Wikipedia passages and injecting known factual errors, and then ran our annotation pipeline on this controlled dataset. Across 100 samples, our annotation pipeline correctly detected 729/904 of the injected errors (80.6% recall), and falsely flagged 15.8% of unchanged entities (i.e., 15.8% false positive rate).

## 3.2 PROBLEM FORMULATION AND TOKEN-LEVEL PROBE TRAINING

**Problem formulation.** We frame hallucination detection as a token-level binary classification task supervised by entity-level annotations. Given a query $\mathbf{q}$ and a chat model $M$, let $\mathbf{t} = (t_1, \ldots, t_n) \sim M(\mathbf{t} \mid \mathbf{q})$ denote the generated response, where each $t_i$ represents a token. For each response $\mathbf{t}$, our annotation pipeline identifies a set of *entity spans* $\mathcal{S}$, where each span $s \in \mathcal{S}$ consists of a sequence of token indices $[s^{\text{start}}, \ldots, s^{\text{end}}]$. Each span has a binary label $y_s \in \{0, 1\}$, where $y_s=1$ denotes a hallucinated entity and $y_s=0$ denotes a supported entity.

While the ground truth is defined at the *entity level*, our goal is to train a detector that operates at the *token level* for real-time monitoring. The detector must assign a probability $p_i \in [0, 1]$ to every token $t_i$, representing the likelihood that $t_i$ is part of a hallucinated entity. For evaluation, we map these token-level probabilities back to the entity level using a max-aggregation function: the score for an

---

[1]Following Wei et al. (2024b), we append the following postamble to each question in order to prompt the model to give a long, detailed completion: "Provide as many specific details and examples as possible (such as names of people, numbers, events, locations, dates, times, etc.)."

[2]We treat entities labeled as either "Not Supported" or "Insufficient Information" as hallucinated. Spans that cannot be confidently mapped back to spans in the original completion are discarded.

entity $s$ is defined as $\max_{i \in s} p_i$. This score is then compared against the ground-truth label $y_s$ to compute metrics such as AUC.

**Probe architecture.** We denote by $M_{\text{probe}}$ the hallucination detector attached to $M$, consisting of a linear value head[3] and, optionally, LoRA adapters inserted into all layers preceding the head. The value head reads hidden states from an intermediate layer $\ell$ of $M$ and outputs token-level probabilities: $p_i = \sigma\big(\mathbf{w}^\top \mathbf{h}_i^{(\ell)} + b\big)$, where $\mathbf{h}_i^{(\ell)}$ is the hidden state of token $t_i$ at layer $\ell$ and $\sigma$ is the logistic sigmoid function. We always train the value head parameters $(\mathbf{w}, b)$; when LoRA adapters are present, we train those as well.

**Span-max loss.** Naively applying binary cross-entropy (BCE) loss between token predictions $p_i$ and span labels $y_s$ (for $i \in s$) can be problematic because annotated spans are often longer than the actual error. For example, in the span "born in 2002," only the final token ("02") may be incorrect. Since hallucination signals concentrate at specific high-information tokens, we combine naive token-wise BCE with a "span-max" loss that applies BCE between the span label $y_s$ and the maximum prediction within that span. For hallucinated spans ($y_s{=}1$), the span-max term rewards the probe if *at least one* token scores high; for supported spans ($y_s{=}0$), it requires *all* tokens to score low. We anneal the weighting between these two loss components during training: early on, the token-wise term provides stable gradients, while later the span-max term sharpens focus on the most informative tokens.

**Regularization.** For LoRA-based probes, we include a regularization term to constrain changes to the model's behavior. We experiment with two regularization strategies: standard language modeling loss (next-token prediction) or KL divergence. This regularization balances detection performance with behavioral preservation, explored in detail in Appendix J.

### 3.3 BASELINES

To contextualize the performance of our probes, we compare against several uncertainty-based metrics. In particular, we evaluate token-level entropy, token-level perplexity, semantic entropy, and a black-box self-evaluation method. See Appendix G for additional details.

- **Token-level entropy:** Uncertainty in the next-token distribution; higher values indicate the model considered many plausible continuations.
- **Token-level perplexity:** How "surprised" the model is by its own token choice; higher values signal lower confidence.
- **Semantic entropy:** Measures uncertainty over *semantic meanings* rather than surface forms via clustering multiple sampled completions (Kuhn et al., 2023; Farquhar et al., 2024). See Section 2 for a description, and Appendix G.2 for implementation details.
- **Black-box self-evaluation:** Prompting the model to judge whether a sentence from its own output contains a hallucination. Full details and results are provided in Appendix G.3.

## 4 LONG-FORM HALLUCINATION DETECTION

### 4.1 EXPERIMENTAL SETUP

**Models.** We primarily focus our analysis on two models ("*primary models*"): Llama-3.1-8B-Instruct and Llama-3.3-70B-Instruct (Grattafiori et al., 2024). We also replicate key results using three additional models ("*secondary models*"): Gemma-2-9B-IT (Riviere et al., 2024), Qwen-2.5-7B-Instruct (Yang et al., 2025), and Mistral-Small-24B-Instruct-2501 (Mistral AI, 2025).[4]

**Training data.** For our primary models, we train on a mixture of long-form and short-form data.[5] For each model $M$, we sample $n_{\text{LF}}^M$ long-form prompts from LongFact and LongFact++, and $n_{\text{SF}}^M$

---

[3]We attach the probe head at layer $\ell = \lfloor 0.95 \times \text{num\_layers} \rfloor$ unless otherwise noted.

[4]All models studied in this paper are instruction-tuned models. For brevity, model names will henceforth exclude the "Instruct" or "IT" suffix.

[5]We find that training on a mix of long-form and short-form data yields the best overall performance; training only on long-form data also works well (see Section 5.1).

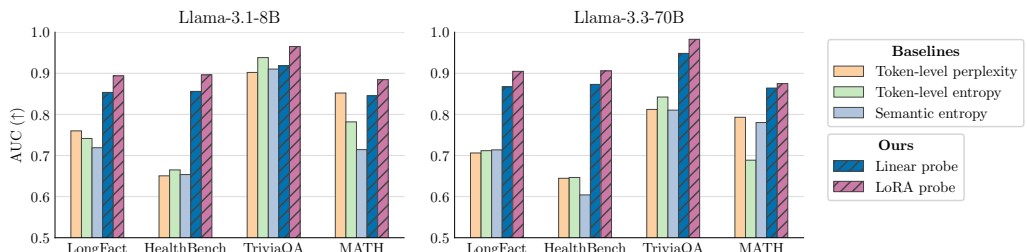

Figure 2: **Token-level probes detect hallucinated entities.** In long-form generation settings (Long-Fact, HealthBench), linear probes far outperform uncertainty-based baselines, with LoRA probes improving performance even further. Our probes also perform well in short-form settings (TriviaQA), and out-of-distribution reasoning domains (MATH). See Appendix H for evaluation on LongFact++ prompts, and results for secondary models.

short-form prompts from TriviaQA (Joshi et al., 2017), and then generate one completion per prompt.[6] Labels for long-form completions follow the pipeline in Section 3. For short-form completions, we extract and label only the single entity span corresponding to the answer of the trivia question (the "*answer entity span*"). For the results in this section, we train probes on our primary models using labeled generations from *all models* (primary and secondary), yielding a training corpus of ∼25,000 total samples. For more details on data generation, see Appendix C.2.

**Evaluation.** Unless otherwise noted, detectors are always evaluated in the same-model setting: each probe is tested on generations from its own original model. All models share a common long-form test set of 1,000 LongFact and 1,000 LongFact++ prompts. We also evaluate performance on short-form completions using TriviaQA (Joshi et al., 2017). To test generalization, we evaluate on two held-out datasets: HealthBench (Arora et al., 2025), which contains unseen long-form medical-domain prompts, and MATH (Hendrycks et al., 2021b), which tests performance on an out-of-distribution mathematical reasoning task without discrete entities.

Our evaluation measures how well each method classifies individual entities as either supported or hallucinated. To do this, we assign a score to each entity using the *span-max* rule, where an entity's score is the maximum of any token within its span. In long-form tasks, we score all annotated entities, while for short-form QA, we score only the single entity corresponding to the answer of the question. For mathematical reasoning, which lacks entities, we score the entire completion by its maximum token score. The performance of this classification task is then measured by the area under the receiver operating characteristic curve (AUC) and recall at a 10% false positive rate (R@0.1). All specific labeling and scoring protocols are detailed in Appendix E.

## 4.2 RESULTS

In long-form settings (LongFact and HealthBench), token-level probes markedly outperform baselines for both primary models (Figure 2; Table 2). Simple linear probes consistently achieve AUCs above 0.85, and LoRA probes improve even further, pushing AUCs above 0.89. In comparison, the uncertainty-based baselines all struggle, failing to exceed 0.76 AUC.

In the short-form setting (TriviaQA), the baselines are stronger than in the long-form setting, yet probes still lead. Our LoRA probes consistently achieve greater than 0.96 AUC, and linear probes also perform well.

Notably, our probes also achieve strong results on the MATH dataset. This out-of-distribution performance suggests our method captures signals of correctness that generalize beyond its original target of fabricated entities. An annotated example from the MATH dataset is provided in Appendix B.2.

We replicate the long-form results on the three secondary models, training each on only 2,000 annotated samples of its own long-form generations. The results are similar: LoRA probes again

---

[6]For Llama-3.1-8B, we use $n_{\text{LF}}^M{=}8{,}000$ and $n_{\text{SF}}^M{=}2{,}000$; for Llama-3.3-70B, we use $n_{\text{LF}}^M{=}8{,}000$ and $n_{\text{SF}}^M{=}1{,}000$. For secondary models (Gemma, Qwen, and Mistral), we use $n_{\text{LF}}^M{=}2{,}000$ and $n_{\text{SF}}^M{=}0$.

outperform linear probes, with AUCs ranging between 0.87–0.90 on LongFact generations. Full results for secondary models are displayed in Table 3.

While LoRA probe AUCs approach or exceed 0.9 in several settings, R@0.1 on long-form tops out around 0.7, i.e., at 10% false positive rate, the detector recovers roughly two-thirds of hallucinated entities. These results underscore both the practical gains over standard uncertainty-based baselines, and also the remaining headroom before such methods can be used broadly in high-stakes contexts.

## 5 ADDITIONAL EXPERIMENTS

### 5.1 GENERALIZATION BETWEEN SHORT- AND LONG-FORM GENERATION SETTINGS

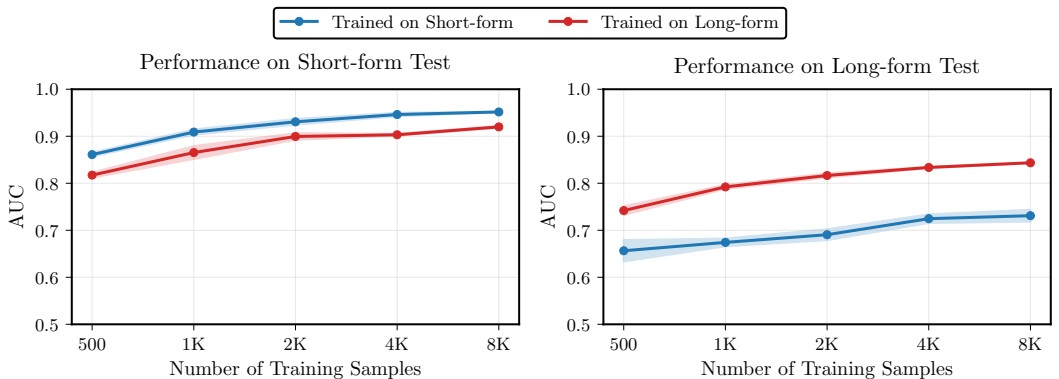

Figure 3: **Generalization between short- and long-form generation settings** (Llama-3.1-8B; 3 seeds per point; mean $\pm$ standard deviation AUC shown). The x-axis refers to the number of training examples from the regime indicated in the legend. *Left:* Performance on the *short-form* (TriviaQA) test set. Blue: probes trained only on short-form. Red: probes trained only on long-form. *Right:* Performance on the *long-form* (LongFact) test set. Performance gaps between training regimes are smaller on short-form tests (<0.05 AUC) but much larger on long-form tests (~0.10 AUC).

Most prior work on hallucination detection focuses on short factoid QA (Orgad et al., 2025; Kossen et al., 2024; Tillman & Mossing, 2025), where labeling is clean and single-answer verification is straightforward, whereas our target use-case is long-form, multi-claim generations. We examine whether token-level hallucination probes trained in one regime (long- vs short-form) generalize to the other. For these experiments we use linear probes rather than LoRA probes, though we expect the qualitative trends to carry over since the asymmetries we observe are driven by data distribution differences rather than probe capacity.

**Long-form training → short-form evaluation.**   We first ask whether probes trained on long-form data generalize to short-form evaluation. Figure 3 (left) confirms that probes trained only on long-form training data (LongFact) achieve high AUC on short-form test data (TriviaQA), with only a small performance gap (<0.05 AUC) compared to short-form-trained probes. The small gap suggests that long-form-trained probes capture broadly transferable cues for factuality, even when evaluated on much shorter, cleaner completions.

**Short-form training → long-form evaluation.**   Motivated by the fact that labeling short-form datasets is far easier and more efficient than annotating long-form content, we next test the reverse: can we train probes *only* on short-form data and have them perform well on long-form hallucination detection? This is an attractive idea in practice; if it worked, one could avoid the high cost of long-form annotation while still solving the harder problem.

The results in Figure 3 (right) show that, although short-form-trained probes do improve with more short-form data, they remain ~0.10 AUC behind long-form-trained probes across all training-set sizes. This gap persists despite the same probe architecture and training procedure, indicating that solving short-form hallucination detection does not automatically yield strong long-form performance.

These asymmetric generalization results highlight the importance of including long-form data in training, especially since long-form, multi-claim outputs are where most real-world hallucinations occur in modern LLM applications.

## 5.2 Cross-model generalization

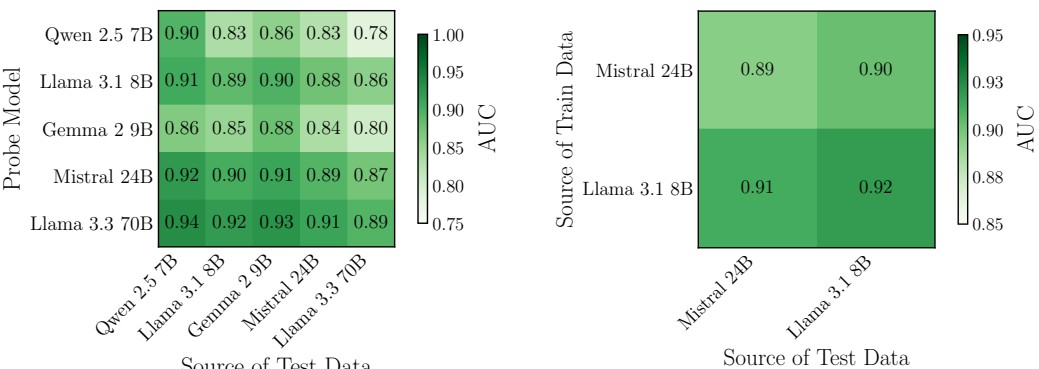

Figure 4: **Hallucination probes exhibit strong cross-model generalization.** *Left*: Cross-model generalization across all five models. The y-axis is the detector model (where the probe was trained), and the x-axis is the test data model (whose generations are evaluated). *Right*: Cross-model training-testing comparison for the Mistral-Small-24B probe. The y-axis indicates which model's generations were used as the training data source, and the x-axis indicates which model's generations were used as the test data source.

An important question is whether our hallucination probes can only identify hallucinated content in their own outputs, or whether they generalize to detecting hallucinations in outputs from *other* models as well. Success in the latter case would indicate that the probe captures fundamental, model-agnostic signals of factuality rather than relying solely on internal signals specific to the generating model. This cross-model analysis addresses two related but distinct questions that are crucial for understanding both the nature of our detection approach and its practical deployment potential.

First, we investigate whether probes trained on one model's generations can effectively detect hallucinations in completions produced by different models—a capability that would enable universal hallucination monitoring across diverse LLM deployments.[7] Second, we examine whether training probes on other models' generations (rather than their own) can still yield effective detectors, which would inform strategies for leveraging high-quality training data from more capable models to supervise smaller or less reliable ones.

**Can probes trained on one model detect hallucinations in other models' outputs?** Following the experimental setup in Section 4.1, we train LoRA-based probes for each of our five models on their own annotated completions and evaluate them on the test sets of the other models. As shown in Figure 4 (left), the results reveal strong cross-model transfer: off-diagonal AUC scores are typically within 0.02–0.04 of the diagonal (same-model performance). This generalization suggests that the probes mostly capture model-agnostic features of factuality, rather than model-specific signals.

The left heatmap additionally highlights two complementary scaling effects. First (row-wise), probes trained on larger models consistently achieve higher performance across all test conditions, suggesting that stronger detectors make better supervisors for other models.[8] Second (column-wise), all probes perform better when evaluating completions from smaller models than from larger ones, consistent with the intuition that larger models may produce factually correct content that smaller detector models simply lack the knowledge to verify, making those cases harder to identify as non-hallucinations.

---

[7]We note that this setting would require passing the generated completions through the monitoring model for analysis, which incurs additional cost compared to token-level streaming detection performed directly on the generating model.

[8]Note that this comparison does not control for the number of probe parameters: with identical LoRA settings, larger models yield more adapter parameters, which may partly explain their higher scores.

**Can probes learn effectively from other models' training data?** Figure 4 (right) shows that the Mistral-Small-24B probe achieves comparable performance when trained on its own data or on Llama-3.1-8B data, with AUC differences within 0.02. This further reinforces the strong transferability observed, extending even to the choice of training data.

### 5.3 HALLUCINATION MONITORING ENABLES SELECTIVE ANSWERING

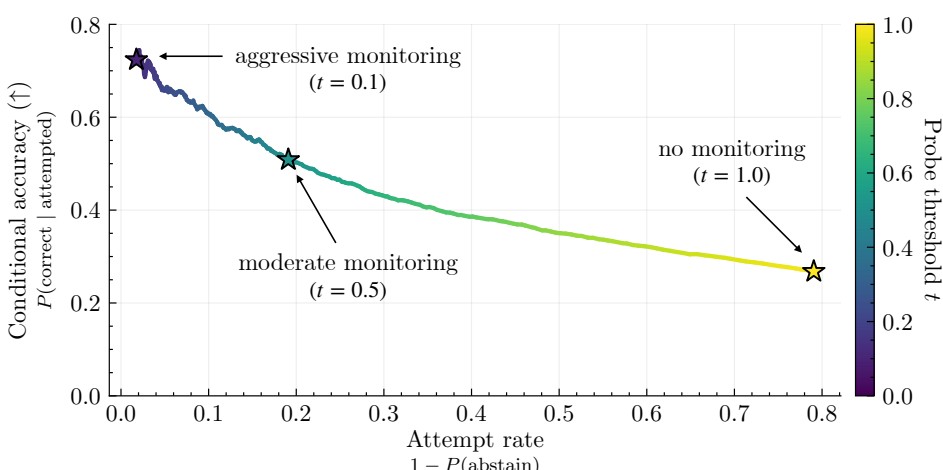

Figure 5: **Real-time hallucination monitoring enables selective answering with higher reliability.** In a QA setting, we monitor probe scores for each token during generation; when any token's probe score exceeds threshold $t$, we halt generation and output an abstention. This yields a system that can *selectively answer* only when the underlying model is confident, achieving higher conditional accuracy, though at the cost of attempting fewer questions. Results for Llama-3.3-70B are displayed.

Beyond detecting hallucinations after generation, our probes enable real-time intervention during generation, opening possibilities for dynamic response modification based on confidence signals. As a proof of concept, we explore one such intervention: selective answering, where the system monitors hallucination signals during generation and abstains when risk exceeds a threshold.

We evaluate this approach on SimpleQA (Wei et al., 2024a), a factual QA benchmark where responses are categorized as correct, incorrect, or not attempted (abstained). Using probes trained with KL regularization ($\lambda_{KL}$=0.5), we monitor each token's probe score during generation. When any token's score exceeds threshold $t$, we halt generation and output an abstention (e.g., "I don't know").

By selectively abstaining on uncertain questions, the system can improve reliability on the questions that it *does* answer. We measure conditional accuracy (accuracy on attempted questions) and attempt rate (fraction of questions attempted), capturing the trade-off between reliability and utility. Figure 5 shows results for Llama-3.3-70B across different probe thresholds. With no monitoring ($t$=1.0), the system attempts a majority of questions (~80%) but achieves low conditional accuracy (<30%). As we enable more aggressive monitoring by lowering the probe threshold, the system becomes increasingly selective, attempting fewer total questions, but with increasing conditional accuracy.

This pattern holds consistently across all models tested: selective answering improves conditional accuracy while reducing attempt rate (see Table 4 in Appendix I).

By monitoring hallucination risk in real-time, we can build systems that better recognize when they should abstain rather than risk providing misinformation—a critical capability for safe deployment in high-stakes applications.

## 6 LIMITATIONS

**Conceptual scope of entity-level detection.** Our approach targets entity-level hallucinations—fabricated names, dates, citations, and similar atomic units—rather than the full spectrum of factual

errors that language models produce. This design choice enables clean token-level supervision and real-time detection, but it also constrains what our method can reliably identify.

The core limitation is that many factual errors are *relational* rather than atomic: they assert incorrect relationships between entities that are themselves real. In such cases, there is no single hallucinated entity to detect. We adopt a flexible operational definition of "entity" (Appendix C.5), which allows the automated annotation pipeline to exercise judgment rather than relying on a rigid taxonomy. A consequence of this flexibility is that the pipeline sometimes labels broader spans—phrases or clauses rather than atomic entities—when the error cannot be localized to a single entity (see Appendix C.4 for further analysis). This captures errors that would otherwise be missed, but it also stretches beyond the atomic entity-level framing and introduces ambiguity about what exactly constitutes an entity.

Beyond relational errors, our method does not explicitly target reasoning errors or context-dependent hallucinations (where generated text contradicts provided source material rather than world knowledge). That said, our strong out-of-distribution performance on MATH suggests that the probes may learn a broader signal of model correctness than the training objective explicitly targets. The extent of this generalization warrants further study.

**Annotation noise.** Our automated pipeline introduces noise into both training and evaluation data. This stems from multiple sources: LLM judges can make errors when verifying facts, search engines may fail to retrieve relevant evidence, and the mapping between complex claims and specific entities is sometimes ambiguous. Our controlled evaluation quantifies this concretely: the pipeline achieves 80.6% recall on synthetic hallucinated entities with a 15.8% false positive rate (Appendix F). Such labeling errors create a performance ceiling that constrains both training effectiveness and evaluation confidence.

**Limited reliability.** While our probes achieve promising AUC scores and outperform baselines, practical reliability remains insufficient for production deployment. Our best LoRA probes achieve only ~70% recall at a 10% false positive rate on long-form text. The selective answering experiments clearly illustrate this gap: meaningfully reducing hallucination rates currently requires sacrificing ~50% of correct answers. This trade-off renders the current approach impractical for real-world scenarios where users expect both accuracy and helpfulness.

## 7 DISCUSSION

This work represents an initial step toward practical, real-time hallucination detection in long-form generation. By framing hallucination detection as a token-level sequence labeling problem, our streaming approach enables monitoring during generation without the computational overhead of external verification pipelines. A key contribution is our automated annotation technique using a frontier LLM augmented with web search to create fine-grained token-level labels that distinguish between grounded and fabricated entities. Our LoRA-based probes achieve 0.89+ AUC on long-form hallucination detection, significantly outperforming uncertainty-based baselines.

While our approach shows promise, important limitations remain before practical deployment is feasible. Our automated annotation pipeline introduces noise that constrains training effectiveness, and current probe performance—achieving only ~70% recall at 10% false positive rate on long-form text—remains insufficient for high-stakes applications. Additionally, our focus on entity-level hallucinations captures only a subset of problematic model outputs.

Looking ahead, future work should address these fundamental gaps through higher-quality annotation techniques, more sophisticated generation-time interventions that preserve informativeness while reducing errors, and expanding detection beyond entity spans to capture reasoning and relational hallucinations. Despite these challenges, our streaming detection approach demonstrates the feasibility of token-level hallucination monitoring and provides a promising foundation for advancing real-time factual reliability in language models.

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

APPENDIX

## M Statement on AI-assisted tool usage 43

## A  CODE AND DATASET AVAILABILITY

Our code and datasets will be released upon publication.

## B  ADDITIONAL ANNOTATED SAMPLES

### B.1  EXAMPLE FROM HEALTHBENCH

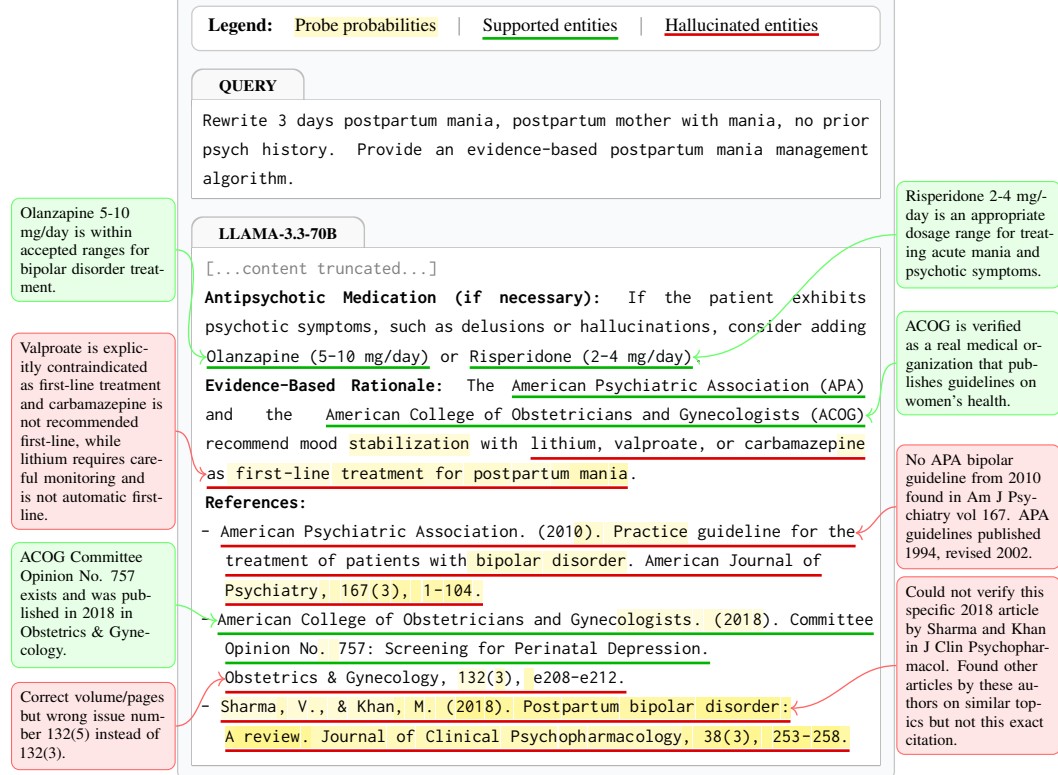

Figure 6: An annotated example of hallucination detection in a response to a HealthBench prompt. The underlines indicate entity spans labeled by our annotation pipeline: green denotes entities labeled as supported, while red denotes entities labeled as hallucinated. Hallucination detection probe scores for each token are shown as yellow highlights, with the intensity reflecting the score's magnitude (scores below 0.25 are not shown).

## B.2 EXAMPLE FROM MATH

---

**Legend:** Probe probabilities

**QUERY**

What is the smallest positive perfect cube that can be written as the sum of three consecutive integers?

**LLAMA-3.1-8B**

To find the smallest positive perfect cube that can be written as the sum of three consecutive integers, we need to consider the properties of perfect cubes and consecutive integers. A perfect cube is a number that can be written as the cube of an integer, i.e., n^3, where n is an integer. Consecutive integers are integers that follow each other in order, i.e., n, n+1, n+2, etc.

Let's consider the smallest perfect cubes and see if we can express them as the sum of three consecutive integers:

1. 1^3 = 1 (1 = 0 + 1 + 0)
2. 2^3 = 8 (8 = 3 + 4 + 1)
3. 3^3 = 27 (27 = 9 + 10 + 8)
4. 4^3 = 64 (64 = 21 + 22 + 21)
5. 5^3 = 125 (125 = 42 + 43 + 40)

We can see that 8 is the smallest perfect cube that can be written as the sum of three consecutive integers.

Therefore, the smallest positive perfect cube that can be written as the sum of three consecutive integers is 8.

---

Figure 7: An annotated example of hallucination detection in a response to a MATH prompt. Hallucination detection probe scores for each token are shown as yellow highlights, with the intensity reflecting the score's magnitude (scores below 0.30 are not shown).

# C  DATASET CONSTRUCTION DETAILS

## C.1  LONGFACT++

While LongFact (Wei et al., 2024b) aims for topical diversity, we observed structural limitations in elicited responses, finding that the prompts often yield vague and generic information. To address these limitations, we developed LongFact++, a dataset 10 times larger than LongFact, with three objectives: (1) increase sample size, (2) diversify query structures to better reflect real user questions, and (3) expand coverage to verifiable fact-rich domains.

Specifically, we construct LongFact++ to consist of:

- **Topic-focused queries**: We first use a frontier LLM (Claude Sonnet) to iteratively generate a list of 1,000 highly specific seed topics spanning law, medicine, the natural sciences, engineering, history, geography, and arts & culture. These specific seeds avoid broad categories (e.g., "medicine") in favor of precise formulations (e.g., "molecular mechanisms of viral DNA replication in herpesviruses"). For each seed topic, a frontier LLM generates 20 diverse questions that vary in length, structure, and focus while remaining grounded in the same seed, yielding natural queries that elicit structurally varied responses.

- **Biography questions**: We include an additional 500 biography-related prompts sourced from (Min et al., 2023), using a fixed prompt template to generate questions about notable individuals.

- **Citation-focused prompts**: We generate an additional ∼1,000 prompts related to various research topics, where we specify to provide references throughout the text, eliciting completions rich in verifiable citation-based entities.

- **Legal prompts**: We add 500 prompts based on well-known legal cases scraped from the Wikipedia page "List of landmark court decisions in the United States" [9], generating queries using predefined prompt templates that ask for the factual background of each case. We found that prompting models with less famous cases resulted in high refusal rates; these refusal responses contain little training signal for hallucination detection.

LongFact++, like LongFact, is a *set of prompts*, and does not itself serve as training data for hallucination detection. We use LongFact and LongFact++ to elicit hallucination-rich responses from target models. For each target model, we sample completions with temperature 0.1 and a maximum generation length of 2,048.

For a subset of questions (specifically for biography questions), we filter out model responses that are explicit refusals.

## C.2  DATASET SPLITS

**Shared long-form test set.** All models use the same 2,000-prompt long-form test set: 1,000 LongFact and 1,000 LongFact++. The LongFact++ portion is sampled uniformly across medical, legal, citations, and biographies to balance domain coverage. These exact 2,000 prompts are identical across models, although their corresponding generations differ.

**Long-form training reservoir and per-model sampling.** After removing the shared test prompts, the remaining LongFact/LongFact++ prompts form a training reservoir of approximately 22,000 prompts. For each model we sample from this pool and generate one completion per prompt (temperature 0.1; max length 2,048). Primary models (Llama-3.1-8B, Llama-3.3-70B) use $n_{\mathrm{LF}}$=8,000 long-form prompts per model; secondary models (Gemma-2-9B, Qwen-2.5-7B, Mistral-Small-24B) use $n_{\mathrm{LF}}$=2,000.

**Short-form (TriviaQA).** For each model, we build a balanced TriviaQA split following the method of Tillman & Mossing (2025): we sample five completions per question at temperature 1.0; auto-judge each against the ground-truth answer with an LLM-as-a-judge; retain only questions that are unanimously correct (5/5) or unanimously incorrect (0/5); balance the resulting dataset. Within any single model, a TriviaQA question appears in either that model's train or test set, but not both. For

---

[9] https://en.wikipedia.org/wiki/List_of_landmark_court_decisions_in_the_United_States

evaluation, each completion contributes a single *answer span* (the specific entity corresponding to the model's answer), which we score via span-max. For Llama-3.1-8B, we generate $n_{SF}=2,000$ responses to short-form questions; for Llama-3.3-70B, we generate $n_{SF}=1,000$.

**HealthBench.** HealthBench contains 5,000 dialogue-based samples between a model and either a layperson or a healthcare professional. We filter the dataset, choosing to exclude samples that are multi-turn, non-English, or that yield explicit refusals. After filtering, we retain ∼2,000 eligible prompt–response pairs per model for Llama-3.1-8B and Llama-3.3-70B. HealthBench samples are used only for evaluation and never included in training.

**Per-model test sizes.** All models share the 1,000 LongFact and 1,000 LongFact++ long-form test set. Additional test set sizes are:

- Llama-3.1-8B: 2,000 samples from TriviaQA; 1,500 samples from HealthBench; 500 samples from MATH.
- Llama-3.3-70B: 1,000 samples from TriviaQA; 1,500 samples from HealthBench; 500 samples from MATH.

TriviaQA and HealthBench are constructed independently per model and are not guaranteed to be disjoint across models (i.e., the same prompt may appear in both models' splits), but within any single model there is no train-test overlap for a given dataset. The subset of MATH is the same one used in Lightman et al. (2024).

## C.3 ANALYSIS OF ENTITY CATEGORIES

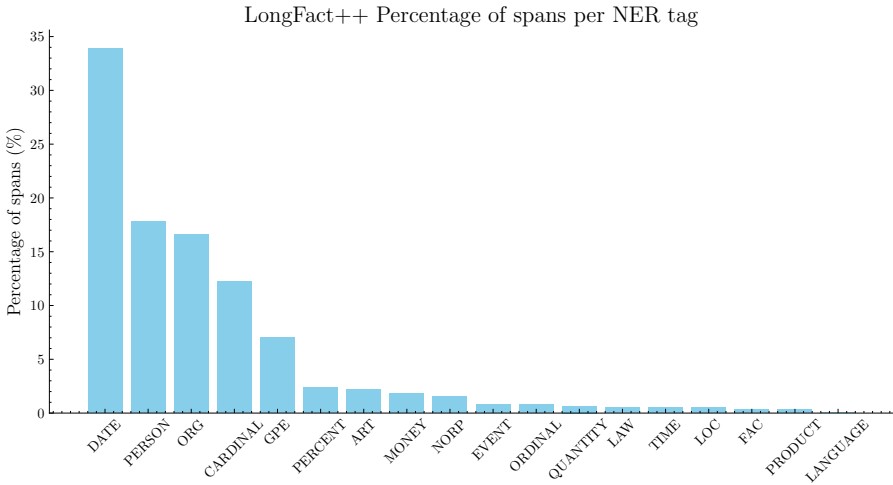

Figure 8: Percentage distribution of named-entity spans in the test set, categorized by the NER (Named Entity Recognition) tags predicted by a spaCy model. Less obvious abbreviations include: ORG (Organization), GPE (Geopolitical Entity), NORP (Nationalities or Religious/Political groups), FAC (Facility), and LOC (Location).

We compute Named Entity Recognition (NER) tags on the test set of Llama-3.1-8B generations on LongFact++ using a spaCy model (en_core_web_sm; Honnibal et al., 2020) and analyze the distribution of entity types. The most common tags correspond to DATE, PERSON, and ORG, which dominate the extracted spans. Mid-frequency categories include CARDINAL, GPE, and PERCENT, while several others appear much less frequently. This distribution highlights a strong skew toward temporal, human-related, and organizational entities in the dataset.

## C.4 ANALYSIS OF ENTITY SPAN LENGTH

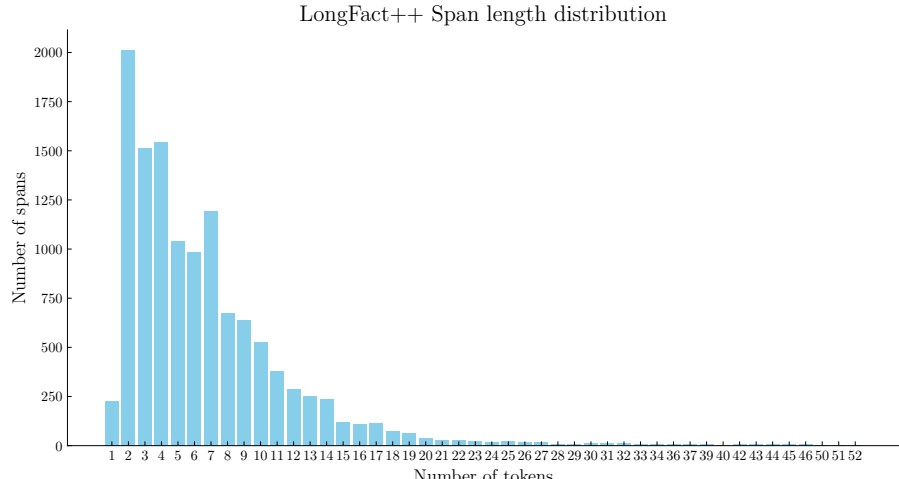

Figure 9: Distribution of entity-span lengths extracted by our automated annotation pipeline from the Llama-3.1-8B test set generations on LongFact++.

We analyze the extracted span lengths in Llama-3.1-8B's LongFact++ dataset. Span length is measured in tokens, and the resulting distribution is shown in Figure 9. The majority of extracted spans are short, with a median span length of 5 tokens and a 90th percentile of 13 tokens. This confirms that the pipeline primarily identifies fine-grained, atomic entities.

However, the distribution also reveals a long tail of longer spans. Inspecting these cases (see Figure 10 for a random sample), we find that longer spans often arise when the pipeline encounters *relational* errors—cases where the hallucination cannot be localized to a single atomic entity. For example, spans like "never ratified by the U.S. Senate due to opposition from Senator Henry Jackson" or "elected to the National Academy of Sciences in 1946" contain real entities (Senator Henry Jackson, National Academy of Sciences) but may assert false relationships. In such cases, the pipeline labels the broader span containing the erroneous claim.

This behavior reflects the trade-off discussed in Section 6: the flexible definition of "entity" allows the pipeline to catch errors that a rigid taxonomy would miss, but it also means that not all labeled spans correspond to atomic, noun-like entities. The span length statistics suggest this is a minority of cases, but it is a meaningful departure from the idealized entity-level framing.

A random selection of 50 entity spans is presented in Figure 10.

> **Random sample of 50 extracted entity spans**
>
> - H3N2 antigenic shift
> - 43.5 meters
> - Bissell & Labarge, 2005
> - Justice Scalia
> - Bachelor's degree in sociology in 1959
> - February 15, 2011
> - Samos, Greece
> - 1948
> - 1667, Leibniz traveled to Paris
> - four children together
> - Over 2,500 citations
> - Journal of the American Ceramic Society, 88(10), 2728-2732
> - Sisters of the Holy Names of Jesus and Mary
> - 6-time NHL All-Star Game selection
> - elected to the National Academy of Sciences in 1946
> - ESPN's ESPY Award for Best Female Athlete
> - 0.5 mg/L
> - HMG-CoA reductase (HMGCR)
> - Martin Luther King Jr.
> - Trans-Pacific Partnership (TPP)
> - never ratified by the U.S. Senate due to opposition from Senator Henry Jackson
> - The Established and the Outsiders
> - Brown v. Board of Education (1954)
> - 1982 to 1994
> - Breakout" (2008)
> - Data Release 1 (DR1): August 2016
> - over 50 patents
> - Fama, 1970
> - 18 CPAs
> - Commissioner v. Glenshaw Glass (1955)
> - 1999
> - Koch's postulates
> - age of 21
> - 1954-1968
> - Psychological Science, 19(11), 1172-1178
> - PRWeek's Agency of the Year (2013, 2015, and 2017)
> - Lindsay Brunnock
> - LISA
> - 15 patients
> - The Pompidou Center
> - Artin & Whaples, 1944
> - 1971
> - 1934-1935
> - uses job costing to track the costs of individual vehicles and process costing to
>   track the costs of production processes
> - September 17, 2010
> - 2006
> - 1978 to 1988
> - since 2013
> - Steve Woolgar
> - August 6, 1991

Figure 10: Random sample of entity spans extracted by our automated annotation pipeline.

## C.5 PROMPT FOR FACT VERIFICATION

---

**System prompt for fact verification**

```
You are a fact-checker. Your task is to fact-check a model's completion to some given
instruction.

Here's the instruction:
<instruction>{instruction}</instruction>

Here's the completion you'll have to fact-check:
<completion>{completion}</completion>

Focus on verifying:
- People (names, affiliations)
- Organizations
- Locations
- Dates and events
- Specific quantities/statistics
- Citations and references

For each entity:
1. Extract the minimal text span (just the entity itself, not surrounding context)
2. Use web search to verify if it's real
3. Label as:
    - "Supported": Verified as correct
    - "Not Supported": Appears to be fabricated or incorrect
    - "Insufficient Information": Cannot verify with available information

Guidelines:
- An entity is correct not only if it is real and it exists (i.e. it isn't fabricated)
but also if it's correct in the context of the sentence and the instruction
- We're interested, not in spotting errors due to small technicalities, but rather in
finding things that the model has clearly bullshitted
- Be skeptical and cautious about highly specific or obscure claims: if you cannot
recall the fact with confidence from your own knowledge or the sources, do not guess;
use "Insufficient Information."
- The extracted span should contain only the specific name, number, citation, etc.
Please do not include anything else within the sentence in the extracted spans
- The spans you extract (the "text" field) should match word-for-word with the original
span in the completion.

Return the output strictly as a JSON array of objects (ordered by the index in which
they appear in the text) following this schema:
```json
[
  {
    "text": "The minimal span containing just the entity (e.g., 'Sarah Chen',
     not 'Dr. Sarah Chen from MIT')",
   "label": "Whether the entity/fact is verified as real, fabricated, or unverifiable",
    "verification_note": "Brief explanation of the verification result"
  },
  ...
]
```
```

---

Figure 11: System prompt used for search-based fact verification (Claude 4 Sonnet).

# D    PROBE TRAINING DETAILS

**Setup.**  Given a query $\mathbf{q}$ and a chat model $M$, the model generates tokens $\mathbf{t} = (t_1, \ldots, t_n) \sim M(\mathbf{t} \mid \mathbf{q})$. Our dataset yields annotations denoting *entity spans* $s = [s^{\text{start}}, s^{\text{end}}]$ (inclusive token indices) with binary labels $y_s \in \{0, 1\}$, where $y_s{=}1$ indicates a hallucinated span. The detector's goal is to assign each token $t_i$ inside labeled spans a probability of being part of a hallucination.

**Probe.**  We denote by $M_{\text{probe}}$ the hallucination detector attached to $M$, consisting of a linear value head and, optionally, LoRA adapters inserted into all layers preceding the head. The value head reads hidden states from an intermediate layer $\ell$ of $M$ and outputs token-level probabilities:

$$p_i = \sigma\big(\mathbf{w}^\top \mathbf{h}_i^{(\ell)} + b\big), \qquad i \in s,$$

where $\mathbf{h}_i^{(\ell)}$ is the hidden state of token $t_i$ at layer $\ell$ and $\sigma$ is the logistic sigmoid function. We always train the value head parameters $(\mathbf{w}, b)$; when LoRA adapters are present, we train those as well. We attach the probe head at layer $\ell = \lfloor 0.95 \times \text{num\_layers} \rfloor$ unless otherwise noted.

**Objective.**  The total loss is a convex combination of a *probe loss*, which trains the hallucination classifier, and a *regularization term*, which constrains changes to the underlying language model:

$$\mathcal{L}_{\text{total}} = (1 - \lambda_{\text{reg}})\, \mathcal{L}_{\text{probe}} + \lambda_{\text{reg}}\, \mathcal{L}_{\text{reg}}, \qquad \lambda_{\text{reg}} \in [0, 1]. \tag{1}$$

The regularizer $\mathcal{L}_{\text{reg}}$ is applied only when training with LoRA; i.e., when the probe is a *linear probe*, $\lambda_{\text{reg}}$ is always zero, as regularization is not needed. We experiment with two losses for the regularization term:

- Language modeling loss ($\mathcal{L}_{\text{LM}}$): standard next-token prediction loss, or
- KL divergence loss ($\mathcal{L}_{\text{KL}}$): KL divergence between the fine-tuned LoRA model and the frozen original model.

These regularization strategies are evaluated in Section J. By default, most experiments use LM regularization with $\lambda_{\text{reg}}{=}0.01$, unless otherwise noted.

**Probe loss: token-wise and span-max.**  We train the probe using binary cross-entropy (BCE) loss between $p_i$ and the label $y_s$ of its containing span $s$. However, annotated spans are often longer than the actual error; for example, in "born in 2002," only the final token ("02") may be incorrect. This creates two challenges: (1) we do not know in advance which tokens within a hallucinated span are incorrect, and (2) hallucination signals are typically concentrated at specific "high-information" tokens (Orgad et al., 2025), not spread uniformly. Optimizing BCE over every token in a span dilutes this signal and risks teaching the probe to activate broadly rather than precisely.

We address this by combining a *token-wise* loss over all tokens with a *span-max* loss over annotated entity spans (Tillman & Mossing, 2025; Sharma et al., 2025). Let $T$ be all token positions and $S$ the set of annotated spans. Define token labels by $y_i{=}y_s$ if $i$ is within an entity span $s$ (an "entity token"), and $y_i{=}0$ if $i$ is outside any entity span (a "background token"). The probe loss is:

$$\mathcal{L}_{\text{probe}} = (1 - \omega) \sum_{i \in T} w_i\, \text{BCE}(y_i, p_i) \; + \; \omega \sum_{s \in S} \text{BCE}\big(y_s, \max_{i \in s} p_i\big), \qquad \omega \in [0, 1]. \tag{2}$$

For a positive label $y_s{=}1$, the max term rewards the probe if *at least one* token in span $s$ scores high; for $y_s{=}0$, it requires *all* tokens within the span to score low. Following Sharma et al. (2025), we anneal $\omega$ from 0 to 1 during training: early on, the token-wise term provides dense, stable gradients; later, the span-max term sharpens the probe's focus on the most informative token in each span (e.g., only the final digits of "born in 2002").

Background tokens greatly outnumber entity tokens, so we up-weight tokens that lie inside any annotated span: $w_i{=}\alpha$ if $i$ is an entity token, else $w_i{=}1$; we use $\alpha{=}10$ unless otherwise noted. This weighting prevents the loss from being dominated by easy background negatives.

# E EVALUATION DETAILS

This section details the specific labeling and scoring methods used to evaluate our probes and baselines across the three distinct task categories.

## E.1 LONG-FORM EVALUATION (LONGFACT, LONGFACT++, AND HEALTHBENCH)

The entity labels for long-form completions are derived directly from our automated annotation pipeline, as described in Section 3. Each entity span is labeled as either supported or hallucinated.

For token-level methods (token-level perplexity, token-level entropy, and token-level probes), we score each entity span using the span-max rule: the span's score is the maximum score of any token it contains. For semantic entropy, the score for an entity span is calculated by taking the text preceding the span as a prefix, sampling $k=10$ continuations, clustering them by semantic equivalence, and computing the entropy of the cluster distribution.

## E.2 SHORT-FORM EVALUATION (TRIVIAQA)

Labels for TriviaQA are created by judging model completions against the known correct answer. Following the method of Tillman & Mossing (2025), we generate five completions for each question and use an LLM-as-a-judge to grade them. We only include questions where the model was unanimously correct or incorrect across all five generations. This binary label is assigned to the single "answer entity span" in the test completion (the particular entity span corresponding to the answer of the question).

Token-level methods are scored using the span-max rule on the single answer entity span. For semantic entropy, we sample $k=10$ full answers to the question. The score is the entropy calculated over semantic clusters of these 10 answers.

## E.3 MATHEMATICAL REASONING EVALUATION (MATH)

The label for each problem in the MATH dataset is determined by the correctness of a single, greedily generated response. We use an LLM-as-a-judge to classify the final numerical or algebraic answer as either correct or incorrect.

As MATH completions lack discrete entities, we adapt our scoring for token-level methods. The score for a generation is the maximum score across all tokens in the entire response. To calculate semantic entropy, we sample $k=10$ completions at temperature 0.6. An LLM then extracts the final answer from each completion, and the score is the entropy computed over semantic clusters of these 10 extracted answers.

# F    LABEL QUALITY VALIDATION

The reliability of our token-level hallucination detection approach fundamentally depends on the quality of our training labels. Since we use an LLM-based annotation pipeline to identify hallucinated entities in long-form text, ensuring high-quality labels is critical for training effective detectors. To validate our dataset quality, we conduct three complementary experiments that assess different aspects of label reliability.

**Addressing annotation hallucinations.**    We face an inherent circularity risk when using LLMs to annotate hallucinations: the annotating LLM could itself hallucinate during the labeling process. This manifests in our pipeline occasionally producing annotations for text spans that do not exist in the original completion. For example, the pipeline might return an annotation claiming the text contains "Accel Partners invested $5 million in Facebook in 2005" and flag the $5 million figure as incorrect, when in reality the original completion never mentioned Accel Partners at all. While these cases are rare, we implement a simple but effective safeguard: all annotated spans must be exactly matched against the original completion text, and any spans that cannot be cross-referenced are automatically discarded. This ensures that hallucinated annotations never contaminate our training data, though it does not guarantee that the labels assigned to valid spans are themselves accurate.

**Human annotation agreement.**    To validate the accuracy of our automated labels, we conduct manual verification on a randomly sampled subset of annotated entity spans. Human annotators independently verify each span without access to the LLM-assigned labels, searching for supporting evidence using web search engines and trusted sources.

On a sample of 50 annotated spans, we find that human annotations agree with the LLM labels in 84% of cases. While this high agreement rate validates our annotation approach, this evaluation method has an important limitation: it only assesses precision (correctness of assigned labels) without measuring recall (proportion of hallucinations detected).

**Controlled evaluation with synthetic hallucinations.**    To rigorously evaluate both precision and recall of our labeling pipeline, we create a controlled test set with known ground-truth hallucinations. Our synthetic evaluation framework operates as follows:

1. **Source selection:** We extract factual content from Wikipedia articles across diverse topics, ensuring high-quality, verifiable source material.

2. **Content transformation:** An LLM rephrases the Wikipedia content into conversational dialogue format while preserving all factual information. This transformation prevents our annotation model from relying on memorized Wikipedia text while maintaining factual accuracy. We acknowledge that this rephrasing step introduces a potential risk of the LLM injecting hallucinations during the transformation. However, in practice we have not observed this risk manifest—we explicitly prompt the LLM to simply rephrase the given content without adding any new factual information, requiring it to express exactly the same facts while only changing the style, order, and format. We believe this rephrasing task is considerably safer than asking an LLM to generate content from scratch, though we acknowledge the theoretical risk.

3. **Controlled hallucination injection:** We prompt an LLM to introduce specific, subtle factual errors into the rephrased content, such as incorrect dates, misattributed quotes, or wrong numerical values. Crucially, we track the exact location and nature of each modification, creating a dataset where we know precisely which spans contain hallucinations.

4. **Pipeline evaluation:** We process these synthetic examples through our standard annotation pipeline and compare the results against ground truth.

Evaluating on 100 synthetic examples containing 904 injected hallucinations, we observe the following performance metrics:

- **Recall:** Our pipeline detects 80.6% (729/904) of the injected hallucinations, indicating that approximately one in five hallucinations may go undetected.

- **Precision on hallucinations:** Not all annotated spans returned by our labeling pipeline necessarily intersect with the spans we purposefully modified to inject hallucinations. However, in cases where an annotated span does coincide with an injected hallucination span, the pipeline assigns the correct label ("Not Supported" or "Insufficient Information") in 100% (729/729) of cases.

- **Precision on factual content:** For spans extracted from unmodified (factual) portions of the text, our pipeline correctly labels them as "Supported" in 84.2% of cases, suggesting a false positive rate of 15.8%.

These results reveal that our labeling pipeline exhibits conservative behavior, with a tendency to over-flag content as potentially hallucinated. While this reduces the risk of training on mislabeled hallucinations, it may also introduce noise by incorrectly flagging some factual content. We note that this evaluation may overestimate real-world performance since Wikipedia-sourced content is likely easier to verify than naturally occurring hallucinations in LLM outputs.

While we would ideally achieve higher recall than 80.6%, we partially address this limitation through our training methodology. In our loss function, we assign significantly higher weight to tokens that coincide with annotated spans compared to the rest of the tokens. This design choice ensures that our probes are not heavily penalized for activating on potentially hallucinated content that our annotation pipeline missed.

**Cross-model annotation robustness.** We compare labels generated by our primary annotator (Claude Sonnet 4) against those from Claude Opus 4 on 224 test completions. Table 1 shows hallucination detection performance when evaluated on test data annotated by each model.

Table 1: Cross-model annotation robustness. Both models are evaluated on 224 completions from Llama-3.1-8B test set, with annotations from either Claude Sonnet 4 or Claude Opus 4.

| Probe model | Annotation model | AUC |
|---|---|---|
| Llama-3.1-8B | Claude Sonnet 4 | 0.9100 |
| Llama-3.1-8B | Claude Opus 4 | 0.9233 |
| Llama-3.3-70B | Claude Sonnet 4 | 0.9330 |
| Llama-3.3-70B | Claude Opus 4 | 0.9406 |

The results demonstrate strong cross-annotator consistency, with only a modest improvement of $\sim$0.01 AUC when using Opus 4 annotations. This suggests that our pipeline produces robust labels that are not overly sensitive to the annotator choice. The slight improvement with Opus 4 may reflect its enhanced capabilities as a more advanced model, potentially offering better judgment in search-based verification tasks and more effective use of search tools.

**Limitations.** While our validation experiments demonstrate satisfactory label quality for training effective hallucination detectors, several limitations warrant discussion. First, our human evaluation sample size is limited due to the time-intensive nature of manual verification. Second, our synthetic hallucination evaluation may not fully capture the complexity of naturally occurring hallucinations, which often involve more subtle forms of factual inconsistency. Finally, our conservative labeling approach, while reducing the risk of false negatives in training data, may limit the ultimate performance ceiling of our detectors.

Despite these limitations, our multi-faceted validation approach provides confidence that our automated labeling pipeline produces training data of sufficient quality for developing token-level hallucination detectors.

# G  BASELINES

## G.1  TOKEN-LEVEL UNCERTAINTY METRICS

**Token-level entropy.**   For token $t_i$ with next-token distribution $p(\cdot \mid \mathbf{q}, \mathbf{t}_{<i})$,

$$H_i = -\sum_{v \in V} p(v \mid \mathbf{q}, \mathbf{t}_{<i}) \log p(v \mid \mathbf{q}, \mathbf{t}_{<i}), \tag{3}$$

where $V$ is the token vocabulary. We compute the maximum-aggregation score over a span $s$ as

$$H_s = \max_{i \in [s^{\text{start}}, s^{\text{end}}]} H_i. \tag{4}$$

**Token-level perplexity.**   For token $t_i$ with next-token distribution $p(\cdot \mid \mathbf{q}, \mathbf{t}_{<i})$,

$$\text{PPL}_i = \exp\big(-\log p(t_i \mid \mathbf{q}, \mathbf{t}_{<i})\big). \tag{5}$$

We compute the maximum-aggregation score over a span $s$ as

$$\text{PPL}_s = \max_{i \in [s^{\text{start}}, s^{\text{end}}]} \text{PPL}_i. \tag{6}$$

## G.2  SEMANTIC ENTROPY

**Semantic entropy.**   *Semantic entropy* (Farquhar et al., 2024) detects hallucinations by measuring uncertainty across semantically equivalent generations. Given a query $\mathbf{q}$, the method samples multiple completions, which are then grouped into clusters $C$ based on semantic equivalence. The probability of a semantic cluster, $p(c|\mathbf{q})$, is operationalized as the fraction of generations in that cluster. Semantic entropy quantifies the uncertainty associated with the distribution $p(c|\mathbf{q})$:

$$H^{SE}(\mathbf{t}, \mathbf{q}) = -\sum_{c \in C} p(c|\mathbf{q})[\log p(c|\mathbf{q})]. \tag{7}$$

**Overview.**   Following Kuhn et al. (2023); Farquhar et al. (2024), we estimate uncertainty over *meanings* by sampling $k$ completions for the same prompt prefix, clustering completions by semantic equivalence, and computing the entropy over cluster probabilities.

**Clustering by semantic equivalence.**   We form clusters via pairwise bidirectional entailment judged by GPT-4.1: two completions $u, v$ are linked if $u \models v$ and $v \models u$. We build an undirected graph on $k$ samples and take connected components as semantic clusters $\mathcal{C} = \{c\}$. Cluster probabilities are empirical frequencies $p(c) = |c|/k$. The semantic entropy is

$$H^{\text{SE}} = -\sum_{c \in \mathcal{C}} p(c) \log p(c). \tag{8}$$

**Task-specific setup.**

- **TriviaQA (short-form):** For each question, we sample $k{=}10$ answers, judge pairwise entailment using only the generated responses, cluster as above, and compute $H^{\text{SE}}$ across these responses.

- **Long-form spans:** For each annotated span $s$, we take the completion prefix up to (but not including) the entity $s$, then sample $k{=}10$ continuations with a target length up to $2\times$ the original span length. We cluster the $k$ continuations and use the resulting $H^{\text{SE}}$ as the span score.

- **Math:** For each question, we generate one greedy completion (temperature 0) and sample $k{=}10$ additional completions at temperature 0.6. We use an LLM to extract the final numerical or algebraic answer from each completion. We compute $H^{\text{SE}}$ by clustering only the $k{=}10$ extracted answers from the temperature-sampled completions based on pairwise entailment.

## G.3 BLACK-BOX SELF-EVALUATION: CAN WE JUST ASK THE MODEL WHETHER IT'S HALLUCINATING?

Given that our white-box probes demonstrate that internal model states contain sufficient information to detect hallucinations, a natural question arises: can we achieve effective hallucination detection by simply asking the model directly, without requiring access to internal representations? This black-box self-evaluation approach would be more practical for deployment scenarios where probe training is infeasible or where models are accessed only through APIs. Here we investigate whether models can reliably identify their own hallucinations when prompted appropriately, and examine how this capability scales from short-form to long-form content.

Our evaluation approach employs a multi-turn conversation format. After the model generates a completion in response to an instruction from our dataset, we select specific sentences within that completion and ask the model to evaluate them in a follow-up question. For sentence selection, we use our existing annotations to identify sentences where either all contained annotated spans are labeled as supported or all are labeled as hallucinated. We discard sentences that contain no annotated spans or have mixed support labels, ensuring clean training signal. We then reference each selected sentence from the model's previous response and ask: "Please evaluate whether the following sentence in our conversation contains a hallucination. Answer with 'Yes' or 'No'." This multi-turn formulation provides the model with the full conversational context necessary for accurate self-evaluation.

We adopt this methodology for several reasons. This black-box self-evaluation approach is highly sensitive to the specific prompt used, and our multi-turn format yielded the best performance after extensive experimentation. Second, it is designed to be feasible for long-form content where sentences often depend on preceding context for proper interpretation—pronouns may lack clear referents, statistics may require earlier context to understand their meaning, and factual claims may build on previously established information. Third, while our dataset consists of span-level annotations, we instead evaluate entire sentences here because: (1) based on our experiments, sentence-level evaluation significantly outperforms direct span-level verification, and (2) it represents a more realistic deployment scenario, since span-level evaluation would require a priori knowledge of which specific text segments to verify.

For comparison with our probe-based approach, we also evaluate our standard LoRA probes on the same sentence-level task by applying the identical sentence selection procedure (sentences containing only supported or only unsupported spans) but treating each complete sentence as a single span rather than individual entities.

Figure 12 presents our results, revealing two key findings. First, the self-evaluation approach shows moderate capability on short-form content (TriviaQA), achieving AUCs between 0.81–0.89. However, its effectiveness does not scale well to long-form content, where performance drops significantly across both models tested (AUCs between 0.58–0.68). Second, we observe that the larger model (Llama-3.3-70B) performs substantially better at self-evaluation than the smaller model (Llama-3.1-8B), suggesting that self-awareness of factual accuracy may improve with model scale.

In addition to prompting-based evaluation, we also implemented a supervised fine-tuning baseline where we trained models specifically for this task using LoRA adapters. We constructed training datasets by pairing each model's completions with multi-turn conversations where the model is asked to evaluate specific sentences, using the same prompting strategy described above. The fine-tuning process optimizes the model to correctly answer "Yes" for sentences containing only unsupported spans and "No" for sentences with only supported spans. Surprisingly, this fine-tuning yields only marginal performance improvements over the prompting approach. We included this baseline for two reasons: to ensure a fair comparison with our trained probes, and as a sanity check to verify that performance limitations were not due to suboptimal prompting or other spurious factors. While we optimized our prompting method before applying fine-tuning, we acknowledge that jointly optimizing prompting strategies with fine-tuning in mind might yield better results. Alternative approaches—such as including reasoning traces in the supervised-fine-tuning data, providing more comprehensive guidelines, or constructing the dataset differently—could potentially improve performance. Nevertheless, we believe our implementation represents a reasonable effort to present this baseline, and we include these results for completeness.

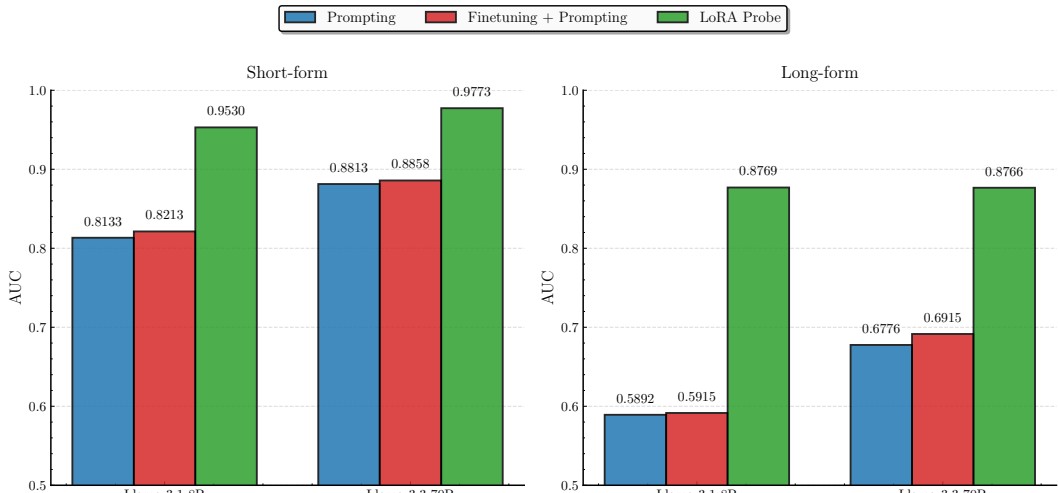

Figure 12: Self-evaluation results comparing AUC performance across datasets. *Left*: TriviaQA (short-form) results where models achieve moderate self-evaluation performance. *Right*: Long-form results demonstrating the performance gap that emerges when scaling to complex, multi-factual content. For long-form evaluation, we train and test on 10,000 samples, while for short-form we use 1,000 samples for Llama-3.1-8B and 2,000 for Llama-3.3-70B. All training and test datasets are balanced between hallucinated and non-hallucinated examples.

While the approach shows some capability on short-form data, the challenge of detecting hallucinations in long-form generations remains substantial. The dramatic performance degradation when moving from TriviaQA to our long-form datasets indicates that the complexity of multi-factual, context-dependent content poses fundamental challenges for self-evaluation approaches. This disparity suggests that self-evaluation faces particular challenges in long-form settings that are better addressed by internal representations.

# H EXTENDED RESULTS: LONG-FORM HALLUCINATION DETECTION

## H.1 RESULTS, PRIMARY MODELS

Table 2: Detection performance on Llama-3.1-8B and Llama-3.3-70B across test sets of LongFact, HealthBench, TriviaQA, and MATH. We report AUC and recall at 10% false positive rate (R@0.1). Probes (linear, LoRA) outperform uncertainty-based baselines; LoRA is strongest across all settings.

| Dataset | Method | Llama-3.1-8B | | Llama-3.3-70B | |
|---|---|---|---|---|---|
| | | AUC ($\uparrow$) | R@0.1 ($\uparrow$) | AUC ($\uparrow$) | R@0.1 ($\uparrow$) |
| **LongFact** | Perplexity | 0.7600 | 0.3616 | 0.7062 | 0.3011 |
| (long-form) | Entropy | 0.7415 | 0.2868 | 0.7118 | 0.3027 |
| | Semantic entropy | 0.7189 | 0.2739 | 0.7138 | 0.3915 |
| | Linear probe | 0.8535 | 0.5878 | 0.8667 | 0.6451 |
| | LoRA probe | **0.8938** | **0.6801** | **0.9048** | **0.7228** |
| **LongFact++** | Semantic entropy | 0.7082 | 0.2368 | 0.6757 | 0.2885 |
| (long-form) | Entropy | 0.7300 | 0.2900 | 0.7389 | 0.3701 |
| | Perplexity | 0.7466 | 0.3400 | 0.7313 | 0.3424 |
| | Linear probe | 0.8678 | 0.6207 | 0.8937 | 0.6971 |
| | LoRA probe | **0.9036** | **0.7052** | **0.9265** | **0.7788** |
| **HealthBench** | Perplexity | 0.6506 | 0.2022 | 0.6446 | 0.2363 |
| (long-form, held-out) | Entropy | 0.6650 | 0.2535 | 0.6466 | 0.2377 |
| | Semantic entropy | 0.6537 | 0.2411 | 0.6042 | 0.2575 |
| | Linear probe | 0.8560 | 0.5843 | 0.8730 | 0.6479 |
| | LoRA probe | **0.8960** | **0.6804** | **0.9057** | **0.7116** |
| **TriviaQA** | Perplexity | 0.9021 | 0.7508 | 0.8121 | 0.5048 |
| (short-form) | Entropy | 0.9382 | 0.8628 | 0.8423 | 0.5524 |
| | Semantic entropy | 0.9103 | 0.7500 | 0.8104 | 0.5525 |
| | Linear probe | 0.9179 | 0.7649 | 0.9484 | 0.8590 |
| | LoRA probe | **0.9651** | **0.9062** | **0.9827** | **0.9486** |
| **MATH** | Perplexity | 0.7143 | 0.1557 | 0.7802 | 0.4299 |
| (reasoning, held-out) | Entropy | 0.7818 | 0.4481 | 0.6887 | 0.3178 |
| | Semantic entropy | 0.8520 | 0.5767 | 0.7930 | 0.3981 |
| | Linear probe | 0.8450 | 0.5739 | 0.8641 | **0.6877** |
| | LoRA probe | **0.8845** | **0.6913** | **0.8750** | 0.6476 |

## H.2 LONGFACT AND LONGFACT++ EVALUATION, SECONDARY MODELS

Table 3: Results for secondary models, as referenced in Section 4.2.

| Dataset | Method | Gemma-2-9B | | Qwen-2.5-7B | | Mistral-Small-24B | |
|---|---|---|---|---|---|---|---|
| | | AUC ($\uparrow$) | R@0.1 ($\uparrow$) | AUC ($\uparrow$) | R@0.1 ($\uparrow$) | AUC ($\uparrow$) | R@0.1 ($\uparrow$) |
| **LongFact** | Linear probe | 0.8200 | 0.5362 | 0.8383 | 0.5432 | 0.8479 | 0.5752 |
| (long-form) | LoRA probe | **0.8733** | **0.6206** | **0.8947** | **0.6645** | **0.8894** | **0.6761** |
| **LongFact++** | Linear probe | 0.8386 | 0.5560 | 0.8467 | 0.5549 | 0.8722 | 0.6278 |
| (long-form) | LoRA probe | **0.8860** | **0.6327** | **0.8961** | **0.6757** | **0.8893** | **0.6927** |

## H.3 PROBE PERFORMANCE PER ENTITY TYPE

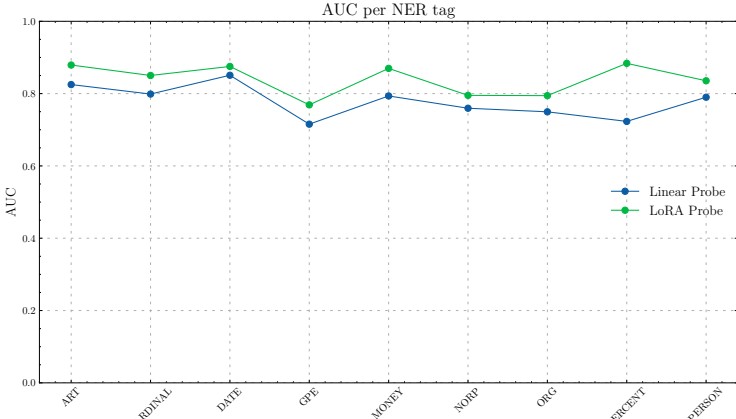

Figure 13: Span-max AUC probe performance by NER tag on the LongFact++ generations of Llama-3.1-8B. Both the linear probe and the LoRA probe show broadly similar performance across entity categories.

We extract Named Entity Recognition (NER) tags by running a pretrained spaCy model (en_core_web_sm; Honnibal et al., 2020) over the LongFact++ generations of Llama-3.1-8B, yielding entity spans together with their predicted NER categories and corresponding hallucination labels. We then evaluate two probes (linear and LoRA) using span-max aggregation strategy. Figure 13 reports AUC scores per NER tag on those categories with at least 200 samples. We observe that performance is broadly similar across categories, without substantial discrepancies. This suggests that probe failures are relatively uniform across entity types.

# I    EXTENDED RESULTS: SELECTIVE ANSWERING

Table 4: Selective answering results for all models. Selective answering (Section 5.3) improves conditional accuracy, at the cost of decreasing the total number of questions attempted. The selective answering results displayed here are obtained using a probe threshold of $t=0.5$.

| Model | Conditional accuracy (%) | | Attempt rate (%) | |
|---|---|---|---|---|
| | No intervention | Selective answering | No intervention | Selective answering |
| Llama-3.1-8B | 19.7 | 48.8 | 10.1 | 2.2 |
| Llama-3.3-70B | 27.9 | 50.4 | 76.1 | 19.1 |
| Mistral-Small-24B | 18.6 | 37.6 | 29.5 | 7.6 |
| Gemma-2-9B | 9.1 | 23.2 | 59.8 | 9.2 |
| Qwen-2.5-7B | 5.5 | 11.3 | 79.4 | 11.2 |

## J    IMPACT ON MODEL OUTPUTS AND BEHAVIOR

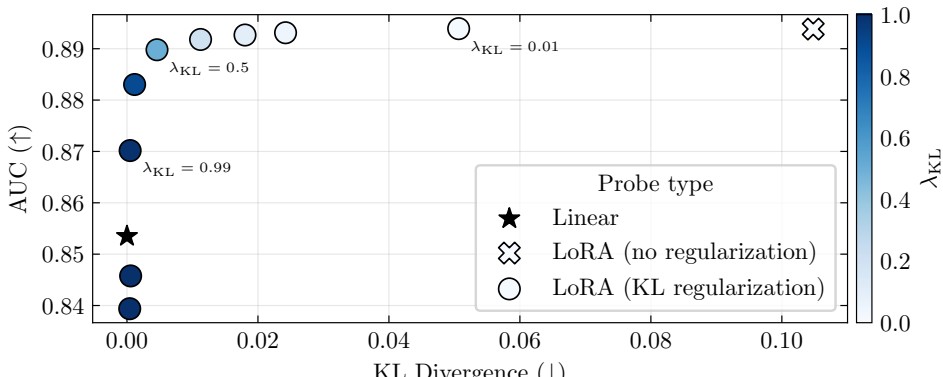

Figure 14: **KL regularization enables tunable detection-preservation trade-offs.** There is a trade-off between hallucination detection performance (AUC) and behavioral preservation (KL divergence) across different probe configurations. KL regularization creates a smooth Pareto frontier as $\lambda_{\mathrm{KL}}$ is varied between 0 and 1, providing tunable control over this trade-off.

Integrating hallucination detection probes directly into the generating model's forward pass offers significant advantages, enabling real-time monitoring while avoiding the computational overhead of external verification. However, this approach introduces an important design consideration: parameter modifications that enhance detection performance may at the same time alter the model's output distribution, potentially affecting generation quality.[10] This creates a spectrum of design trade-offs. At one extreme, linear probes preserve model behavior perfectly by leaving model parameters unchanged, but achieve limited detection performance. At the other extreme, unregularized LoRA-based probes maximize detection accuracy but may significantly alter the model's output distribution.

To evaluate these trade-offs, we measure three aspects of model behavior preservation, alongside detection performance. For model behavior, we assess: (1) KL divergence between the original and modified output distributions; (2) win rate against the original model on Arena-Hard-Auto (Li et al., 2024) as judged by GPT-4.1, measuring overall generation quality; and (3) accuracy on MMLU (Hendrycks et al., 2021a), measuring retention of knowledge and reasoning capabilities. For detection performance, we measure AUC on the LongFact test set.

To directly minimize behavioral changes while preserving detection performance, we employ KL divergence regularization during LoRA training (Section 3). This approach explicitly penalizes deviation from the original model's output distribution, directly targeting the quantity we care about—distribution shift—rather than using proxies like language modeling loss.

Figure 14 illustrates the fundamental trade-off between detection performance and behavioral preservation for Llama-3.1-8B. As we increase the KL regularization strength ($\lambda_{\mathrm{KL}}$), KL divergence decreases (better behavior preservation) while detection AUC slightly decreases, creating a smooth Pareto frontier. KL regularization enables effective navigation of this trade-off space, achieving points with high detection performance and minimal distributional shift. In contrast, unregularized LoRA (cross symbol) achieves high detection performance but with substantial behavioral changes, while linear probes (star) preserve behavior perfectly but limit detection capability.

Table 5 provides a broader evaluation across all approaches. KL regularization at $\lambda_{\mathrm{KL}}=0.50$ achieves good overall balance: near-zero KL divergence (0.0046), a win rate that slightly exceeds the original model (52.8%), preserved MMLU performance (71.2%), while maintaining strong detection performance (0.8898 AUC). This outperforms both unregularized LoRA and LM regularization approaches.

---

[10]Interestingly, we anecdotally find that some LoRA configurations with minimal regularization lead to increased epistemic caution in generations, where models become more likely to acknowledge uncertainty rather than confidently hallucinating. See Appendix J.2 for further discussion and examples.

Table 5: Comparison of model output stability and hallucination detection performance across different probe configurations for Llama-3.1-8B. Win rate estimates have 90% confidence intervals within $\pm 2.1\%$, and all MMLU scores have standard errors of $\pm 0.4\%$.

| | Model performance | | | Probe performance |
|---|---|---|---|---|
| **Configuration** | **KL div.** ($\downarrow$) | **Win rate** (%) ($\uparrow$) | **MMLU** (%) ($\uparrow$) | **AUC** ($\uparrow$) |
| Baseline (linear probe) | 0.0000 | 50.0 | 70.9 | 0.8535 |
| LoRA (no regularization) | 0.1048 | 35.9 | 63.4 | 0.8938 |
| LoRA ($\lambda_{\mathrm{LM}} = 0.01$) | 0.0502 | 34.4 | 67.4 | 0.8938 |
| LoRA ($\lambda_{\mathrm{LM}} = 0.50$) | 0.0610 | 47.2 | **72.1** | 0.8880 |
| LoRA ($\lambda_{\mathrm{KL}} = 0.01$) | 0.0506 | 32.5 | 67.6 | **0.8939** |
| LoRA ($\lambda_{\mathrm{KL}} = 0.50$) | **0.0046** | **52.8** | 71.2 | 0.8898 |

Based on these results, we recommend KL regularization to be used in practice. The $\lambda_{\mathrm{KL}}$ hyperparameter allows practitioners to navigate the detection-preservation trade-off according to their specific deployment context, prioritizing either higher detection performance or closer alignment to original model behavior.

## J.1 Quantitative Analysis

We evaluate the impact of LoRA fine-tuning on model outputs using three complementary metrics:

- **KL divergence** quantifies distributional changes by computing the average KL divergence between the original model ($\pi_{\mathrm{ref}}$) and the LoRA-adapted model ($\pi_\theta$) across token positions: $\mathcal{L}_{\mathrm{KL}} = \frac{1}{T}\sum_{t=1}^{T} D_{\mathrm{KL}}\big(\pi_\theta(\cdot|q,t)\,||\,\pi_{\mathrm{ref}}(\cdot|q,t)\big)$. We generate completions from the original model on 750 prompts from Arena-Hard-Auto (Li et al., 2024), and, over these completions, compute the average token-wise KL divergence between the original model distribution and the modified model distribution.

- **Win rate** measures generation quality via GPT-4.1 pairwise comparisons on Arena-Hard-Auto (Li et al., 2024), with mean and confidence intervals obtained from bootstrap resampling.

- **MMLU accuracy** (Hendrycks et al., 2021a) evaluates knowledge retention using standard zero-shot chain-of-thought prompting. We use Inspect (AI Security Institute, 2024) to run evaluations.

Table 6 provides comprehensive win-rate results across different regularization strengths, showing that regularization values $\lambda$ of 0.50 or higher tend to preserve model quality.

Figure 15 demonstrates that KL-regularized probes achieve superior trade-offs compared to LM-regularized probes. KL regularization creates smooth, predictable behavior as $\lambda_{\mathrm{KL}}$ increases, while LM regularization exhibits erratic patterns—higher $\lambda_{\mathrm{LM}}$ does not consistently reduce KL divergence and can even increase it through overfitting.

Table 6: Win rates on Arena-Hard-Auto for Llama-3.1-8B variants, as judged by GPT-4.1. Each win rate is the mean estimate from a bootstrap analysis (100 resamples of the battle outcomes). The CI represents the corresponding 90% percentile confidence interval.

| Variant | $\lambda$ | Win rate (%) | CI (%) |
|---|---|---|---|
| Baseline | – | 50.0 | $(-0.0\, /\, +0.0)$ |
| LoRA $\lambda_{\mathrm{LM}}$ | | | |
| | 0.01 | 34.4 | $(-2.0\, /\, +2.1)$ |
| | 0.05 | 39.0 | $(-1.8\, /\, +1.7)$ |
| | 0.10 | 43.3 | $(-1.9\, /\, +1.9)$ |
| | 0.20 | 42.0 | $(-1.8\, /\, +2.0)$ |
| | 0.50 | 47.2 | $(-1.8\, /\, +1.5)$ |
| | 0.90 | 48.3 | $(-2.0\, /\, +2.4)$ |
| | 0.99 | 50.4 | $(-2.2\, /\, +2.4)$ |
| | 0.999 | 48.7 | $(-2.1\, /\, +2.1)$ |
| | 0.9999 | 48.2 | $(-1.9\, /\, +1.9)$ |
| LoRA $\lambda_{\mathrm{KL}}$ | | | |
| | 0.00 | 35.9 | $(-2.1\, /\, +2.2)$ |
| | 0.01 | 32.5 | $(-1.9\, /\, +1.8)$ |
| | 0.05 | 39.7 | $(-2.0\, /\, +2.0)$ |
| | 0.10 | 45.3 | $(-1.9\, /\, +2.1)$ |
| | 0.20 | 44.7 | $(-2.3\, /\, +2.2)$ |
| | 0.50 | 52.8 | $(-2.0\, /\, +2.1)$ |
| | 0.90 | 53.3 | $(-1.7\, /\, +1.8)$ |
| | 0.99 | 52.4 | $(-2.1\, /\, +2.0)$ |
| | 0.999 | 48.9 | $(-1.7\, /\, +1.7)$ |
| | 0.9999 | 46.3 | $(-2.0\, /\, +2.5)$ |

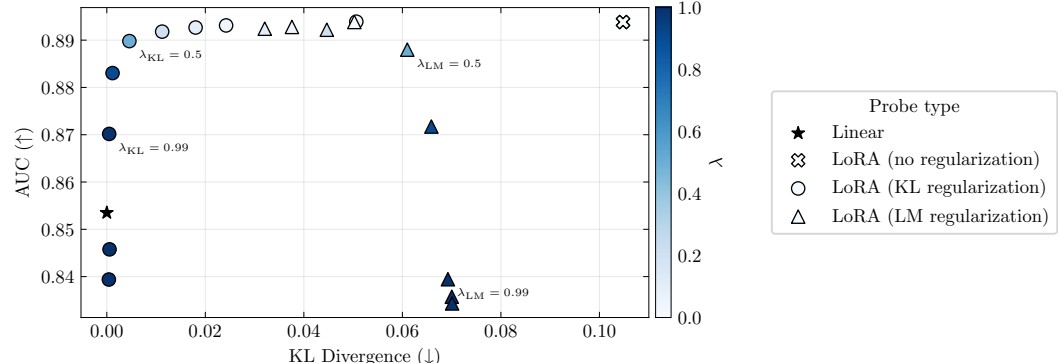

Figure 15: Trade-off between hallucination detection (AUC) and distributional shift (KL divergence). Each point corresponds to a different regularization strength $\lambda$.

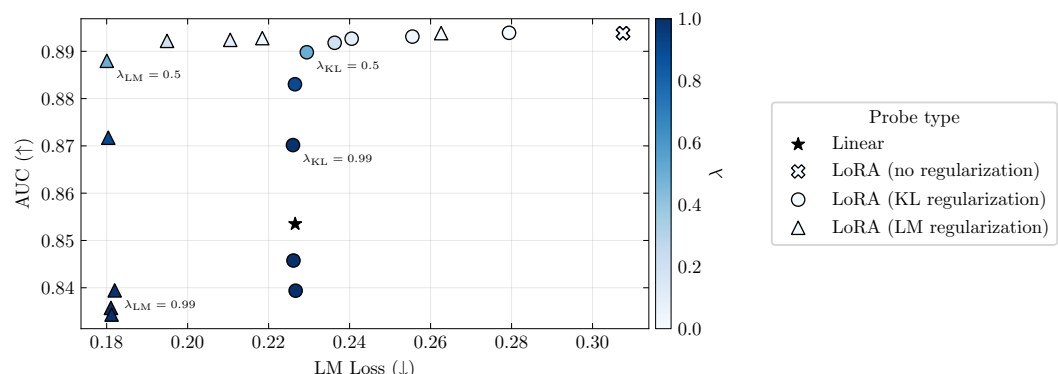

Figure 16: Trade-off between hallucination detection (AUC) and LM loss on model generations. However, LM loss is not ultimately the metric we care about. Minimizing LM loss can result in overfitting and distribution shift.

## J.2 QUALITATIVE ANALYSIS

After training LoRA probes with minimal regularization, we anecdotally observe changes in the model's output distribution that suggest increased epistemic caution and reduced propensity for hallucination.

In this section, we present several anecdotal examples comparing outputs from baseline models against models augmented with LoRA adapters. The LoRA adapters presented here were trained with $\lambda_{LM}$=0.01, with a linear probe head at the final layer (i.e., at the residual stream immediately before the unembedding).[11] All generations are produced with temperature 0. The examples suggest three key behavioral changes: (1) the models become more conservative in making specific factual claims, (2) they more readily acknowledge uncertainty or inability to recall specific details, and (3) in some cases, they explicitly recognize when they might be generating unreliable information.

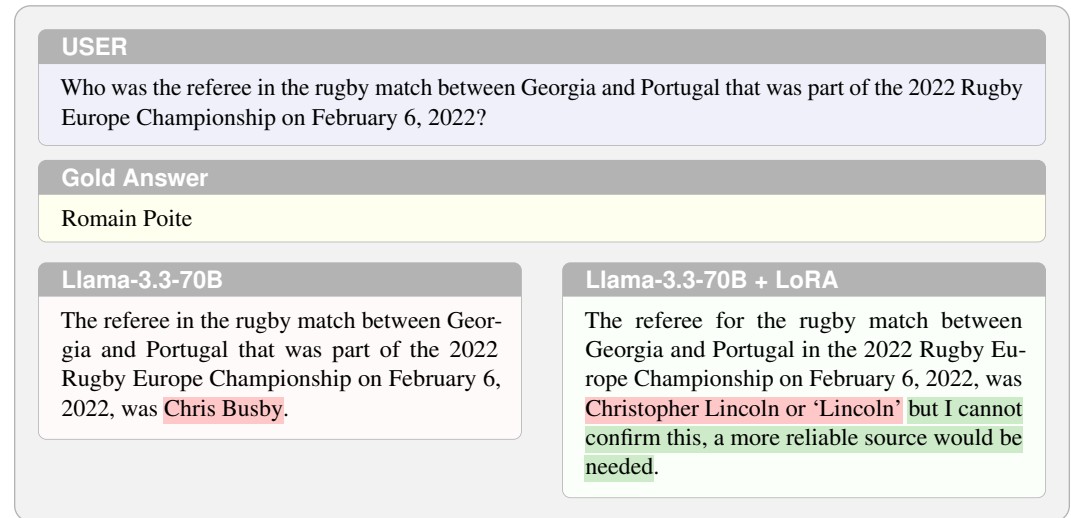

**USER**

Who was the referee in the rugby match between Georgia and Portugal that was part of the 2022 Rugby Europe Championship on February 6, 2022?

**Gold Answer**

Romain Poite

**Llama-3.3-70B**

The referee in the rugby match between Georgia and Portugal that was part of the 2022 Rugby Europe Championship on February 6, 2022, was Chris Busby.

**Llama-3.3-70B + LoRA**

The referee for the rugby match between Georgia and Portugal in the 2022 Rugby Europe Championship on February 6, 2022, was Christopher Lincoln or 'Lincoln' but I cannot confirm this, a more reliable source would be needed.

Figure 17: Example of hallucination detection affecting generation behavior. The baseline Llama-3.3-70B confidently states an incorrect referee name. The LoRA-augmented model exhibits an interesting behavior: it still provides an incorrect answer but immediately acknowledges its uncertainty with "but I cannot confirm this, a more reliable source would be needed."

---

[11]Note that this is distinct from the layer selection used in the rest of the paper, where we attach the probe head to $\lfloor 0.95 \times \text{num\_layers} \rfloor$. Empirically, we only observe these qualitative behavioral changes when optimizing probes on the last layer.

**USER**

O UGLY NINE is an anagram of which singer songwriter?

**Gold Answer**

Neil Young

**Llama-3.1-8B**

The anagram "O UGLY NINE" can be rearranged to spell "Lynyrd Skynyrd", which is a famous American rock band.

**Llama-3.1-8B + LoRA**

I couldn't find any information on a well-known singer or songwriter with the anagram "O UGLY NINE".

Figure 18: The baseline model confidently provides an incorrect answer ("Lynyrd Skynyrd" is not even a valid anagram of "O UGLY NINE"). The LoRA-augmented model correctly expresses inability to solve the anagram, rather than guessing an incorrect answer.

**USER**

Provide a treatment plan for a 29-year-old woman presenting 3 days postpartum with manic symptoms including decreased sleep, elevated mood, and excessive energy. She has a past history of bipolar I disorder. Keep the answer concise (1-2 paragraphs long), specific and evidence-based.

**Llama-3.3-70B**

Given the patient's history of bipolar I disorder and current manic symptoms postpartum, a treatment plan should prioritize mood stabilization and safety. The patient should be started on a mood stabilizer, such as lithium (600-900 mg/day) or valproate (500-1000 mg/day), with close monitoring of serum levels and renal function. Additionally, a second-generation antipsychotic, such as olanzapine (5-10 mg/day) or quetiapine (100-200 mg/day), may be added to help manage manic symptoms and promote sleep.

It is essential to consider the patient's postpartum status and potential breastfeeding plans when selecting medications. Lithium and valproate are generally considered compatible with breastfeeding, but serum levels should be monitored in the infant. Olanzapine and quetiapine are also considered relatively safe during breastfeeding, but the patient should be informed of the potential risks and benefits [...]

**Llama-3.3-70B + LoRA**

Given the patient's history of bipolar I disorder and presentation of manic symptoms 3 days postpartum, a comprehensive treatment plan is crucial. The plan should prioritize mood stabilization and address the patient's decreased sleep, elevated mood, and excessive energy. Medications such as mood stabilizers (e.g., lithium, valproate) or atypical antipsychotics (e.g., olanzapine, risperidone) may be considered. However, it's essential to consider the postpartum context and potential breastfeeding plans when selecting medications. For example, some medications like lithium and valproate are generally used with caution in breastfeeding due to their potential to pass into breast milk.

A treatment plan for this patient may involve hospitalization for close monitoring, especially if there are concerns about the patient's or her baby's safety. Medications like olanzapine or risperidone could be initiated, considering their efficacy in managing manic symptoms. [...]

Figure 19: The baseline generation contains potentially life-threatening errors: the lithium dosing (600-900 mg/day) is underdosed for acute mania, and the claim that lithium is "generally considered compatible with breastfeeding" overstates its safety profile. The LoRA-augmented model is appropriately cautious about breastfeeding risks with lithium and provides safer, albeit less detailed, guidance.

**USER**

collagen for knee arthritis evidence

**Llama-3.1-8B**

Collagen supplements have gained popularity in recent years as a potential treatment for knee arthritis. While some studies suggest that collagen may help alleviate symptoms of knee arthritis, the evidence is not yet conclusive. Here are some key findings from recent studies:

Positive studies:

1. A 2017 study published in the Journal of Medicinal Food found that collagen hydrolysate supplementation (10g/day for 6 months) improved knee pain and function in patients with knee osteoarthritis.

2. A 2019 study published in the Journal of the International Society of Sports Nutrition found that collagen peptide supplementation (10g/day for 24 weeks) improved knee pain and function in athletes with knee osteoarthritis.

3. A 2020 study published in the Journal of Orthopaedic Research found that collagen hydrolysate supplementation (10g/day for 12 weeks) reduced knee pain and improved function in patients with knee osteoarthritis.

[...]

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

I apologize for any confusion caused by the lack of specific references earlier. If you need more information or specific references, I can try to provide them.

Figure 20: The baseline model confidently cites multiple specific studies with full details. The LoRA model initially cites studies but then acknowledges it cannot verify these citations when attempting to provide references.

## K  USE OF EXISTING ASSETS

Table 7: List of models used in this work.

| Name | Source | License |
|---|---|---|
| Llama-3.1-8B-Instruct | Grattafiori et al. (2024) | Meta Llama 3.1 Community License |
| Llama-3.3-70B-Instruct | Grattafiori et al. (2024) | Meta Llama 3.3 Community License |
| Qwen2.5-7B-Instruct | Yang et al. (2025) | Apache License 2.0 |
| Gemma-2-9B-IT | Riviere et al. (2024) | Gemma License (commercial-friendly terms of use) |
| Mistral-Small-24B-Instruct-2501 | (Mistral AI, 2025) | Apache License 2.0 |

Table 8: List of datasets used in this work.

| Dataset | Source | License |
|---|---|---|
| LongFact | Wei et al. (2024b) | Apache License 2.0 |
| TriviaQA | Joshi et al. (2017) | Apache License 2.0 |
| HealthBench | Arora et al. (2025) | MIT License |
| SimpleQA | Wei et al. (2024a) | MIT License |
| MATH | Hendrycks et al. (2021b) | MIT License |

## L  COMPUTE STATEMENT

Training a LoRA-based probe for Llama-3.1-8B-Instruct on the full annotated dataset (as specified in Section 4.1) with a batch size of 8 takes less than 2 hours on an H100 GPU.

We use Claude 4 Sonnet with web search to construct the annotated dataset. Each sample (consisting of roughly 1-2 thousand tokens of input text) costs approximately $0.70 to annotate, and takes approximately 45 seconds (although we run annotations in parallel, since each sample is independent).

## M  STATEMENT ON AI-ASSISTED TOOL USAGE

This work was enhanced through the use of AI-based tools, including ChatGPT, Claude, and various models within the Cursor IDE. These tools were employed to refine writing and assist in code development. Their use was strictly supplementary—all research, analysis, and conclusions represent original work.

