# OpenReview forum: "Real-Time Detection of Hallucinated Entities in Long-Form Generation"
_ICLR.cc/2026/Conference — Submitted to ICLR 2026_

### Official Review · Reviewer_QrDD · 2025-10-14

**Soundness:** 2
**Presentation:** 3
**Contribution:** 2
**Rating:** 4
**Confidence:** 5

**Summary:**

This paper focused on entity-level hallucination detection in long-form generations. The authors introduced an annotation pipeline to automatically annotate hallucinations at the entity level. The authors then proposed to detect hallucinations by training a probing model on the LLM's internal states with a designed span-max loss. The authors conducted experiments on four datasets and with two LLMs, showing that their proposed linear probe achieves the highest performance. Additional analysis also showed the importance of training the linear probe on datasets with long-form generations, highlighting the value of their annotated dataset.

**Strengths:**

1. The writing is clear and easy to follow.
2. The proposed entity-level hallucination detection dataset is valuable and can contribute to future research.
3. The authors conducted extensive experiments and analysis in different settings.

**Weaknesses:**

1. **The novelty of the annotation pipeline.** The annotation pipeline is highly similar to the one proposed in [1], while the authors did not properly cite and discuss their paper. This greatly questions the novelty of this paper as the annotation pipeline and the proposed dataset are two major contributions of this paper.
2. **Task definition.** The term "entity" in the paper is not clearly defined. The authors only list some examples (e.g., named people, organizations, locations, dates, and citations). It is unclear to me what would be considered an entity and what would not, and whether some essential types of entity are overlooked. The authors should provide more details of explanation or statistics of the entity type.
3. **Reliability of annotation.** The authors mentioned that they manually verify the quality of entity labels (Line 184). However, the assessment of the reliability of entity extraction is missing in the verification process. To me, entity extraction is less reliable and may be subjective, so it is important to report metrics like precision or recall rate to show that whether the extraction matches the definition of entity.
4. **Baselines.** Some novel token-level hallucination detection baselines were excluded from the experiments, such as [2] and [3]. The authors should include them to strengthen their claim: "token-level probes markedly outperform baselines" (Line 294)
5. **Lack of deep analysis.** This paper introduced an entity-level dataset, while the experiments only focus on performance comparison without a deeper analysis on how the performance varies across different entities or what entities are easier to be hallucinated by LLMs. Therefore, the contribution of this paper is limited. A deeper analysis could help future research develop better hallucination detection/mitigation approaches at the entity level.

[1]: HalluEntity: Benchmarking and Understanding Entity-Level Hallucination Detection (2025)

[2]: Enhancing Uncertainty-Based Hallucination Detection with Stronger Focus (2023)

[3]: Shifting Attention to Relevance: Towards the Predictive Uncertainty Quantification of Free-Form Large Language Models (2024)

**Questions:**

1. It would be clearer to consistently use either token-level or entity-level within the whole paper.
2. As shown in [4], there may be sentence- and relation-level hallucinations. How does the proposed pipeline handle these types of hallucination?

[4]: Fine-grained Hallucination Detection and Editing for Language Models (2024)

---

> ### Author Response · Authors · 2025-11-23
>
> We thank the reviewer for their detailed and expert feedback. We particularly appreciate the pointers to recent relevant literature (in particular HalluEntity), which we will incorporate into our revised manuscript.
>
> ---
>
> **Addressing weaknesses**
>
> > *The novelty of the annotation pipeline. … [1]: HalluEntity (2025)*
>
> We sincerely thank the reviewer for pointing us to HalluEntity (2025). We were not previously aware of this work, and agree that it is extremely relevant to our work. We have updated our manuscript to highlight the contributions of HalluEntity. Specifically, we added two citations of HalluEntity to our introduction; and have additionally added a whole paragraph to our related work. We hope that these changes adequately highlight and credit HalluEntity with the valuable contribution of introducing the entity-level hallucination detection problem.
>
> That said, we would like to highlight some key differences between our work and that of HalluEntity:
> - Our labeled dataset is orders of magnitude larger than that of HalluEntity. HalluEntity contains 157 text completions, while our labeled dataset contains over 25,000 annotated model completions. The scale of our dataset enables probe training, as a significant number of samples are required for good probing performance (see Figure 4). Additionally, HalluEntity is limited in terms of domain: it consists only of biographies (all prompts are of the format “Question: Tell me a bio of [John Doe].”), whereas our dataset is extremely diverse, furthering its utility for training-based approaches.
>   - We will open-source our annotated dataset upon publication, and hope that it can be an impactful contribution to the community.
> - Our pipeline is substantially simplified compared to that of HalluEntity. The pipeline in HalluEntity extracts atomic claims from the original text, assesses each claim, and then links the claims back to corresponding text spans. We significantly simplify this process, showing that a frontier LLM equipped with web search is capable of directly annotating hallucinated (or non-hallucinated) entity spans.
> - We contribute and evaluate a methodology that is enabled by large-scale data annotation, namely that of probes (both simple linear probes and LoRA probes).
> - Another difference between our dataset and HalluEntity is that HalluEntity does not distinguish between non-hallucinated entities and non-entities. Our fine-grained entity annotations facilitate the possibility of constructing training losses that focus exclusively on entity spans (e.g., training a probe to classify hallucinated vs non-hallucinated entity spans, while not specifying behavior on non-entity spans).
>
> > *Task definition. The term "entity" in the paper is not clearly defined. ... The authors should provide more details of explanation or statistics of the entity type.*
>
> In the manuscript, we define the scope of “entities” through specific examples (subsection 3.1: “e.g., named people, organizations, locations, dates, and citations”), and also provide samples of annotated rollouts in Figure 1 and 6 to give a sense of what spans count as entities. We also provide the exact prompt used to instruct Claude to parse entities in Appendix C.4; the definition there is also given through examples of categories. This relatively flexible definition allows the LLM labeler to make judgement calls, and to label spans that are important to factuality without overfitting to some strict definition of “entity”; Additionally, as frontier LLMs equipped with tool calls continue to improve, we anticipate that the quality of labels that result from this flexible process will also improve.
>
> We appreciate your suggestion to do a quantitative breakdown of entity type, and think this is an important detail to include. Following this suggestion, we performed an analysis of our annotated entities using spaCy NER (Named Entity Recognition) and added the results to appendix (see Appendix C.3).
>
> Count of Entities per NER Tag (LongFact++ test set):
>
> |Category|Count|Percentage        |
> |--------|-----|------------------|
> |ART     |293  |2.18|
> |CARDINAL|1639 |12.20|
> |DATE    |4557 |33.91 |
> |GPE     |946  |7.04|
> |MONEY   |240  |1.79|
> |NORP    |211  |1.57|
> |ORG     |2229 |16.59 |
> |PERCENT |320  |2.38|
> |PERSON  |2395 |17.82|
> |Other   |608  |4.52|

---

> > ### Author Response · Authors · 2025-11-23
> >
> > > *Reliability of annotation. The authors mentioned that they manually verify the quality of entity labels (Line 184). However, the assessment of the reliability of entity extraction is missing in the verification process. To me, entity extraction is less reliable and may be subjective, so it is important to report metrics like precision or recall rate to show that whether the extraction matches the definition of entity.*
> >
> > We thank the reviewer for highlighting the importance of evaluating the extraction step. The Controlled Evaluation (Appendix F) allows us to quantitatively measure the reliability of extraction, not just verification. In this experiment, we injected 904 specific factual errors (hallucinations) into entity spans (e.g., dates, names). We then ran the full pipeline, including both entity extraction and entity classification (as hallucinated or not). Our results allow us to isolate extraction performance. Overall, the recall was 80.6% (729/904 hallucinations detected), and the conditional verification accuracy was 100% (729/729) (conditional verification is whether the hallucination was classified correctly as a hallucination, given that it was extracted as an entity). Whenever our pipeline successfully extracted a span corresponding to an injected error, it correctly labeled it as "Not Supported." This implies that the missing 19.4% of cases were due to extraction failures - i.e., the pipeline failed to correctly extract the entity. Thus, we can report that our extraction recall on hallucinated entities is 80.6%. This confirms that the pipeline reliably extracts a vast majority of entities, although still far from perfect.

---

> > > ### Author Response · Authors · 2025-11-23
> > >
> > > > *Baselines. Some novel token-level hallucination detection baselines were excluded from the experiments, such as [2] and [3]. The authors should include them to strengthen their claim: "token-level probes markedly outperform baselines" (Line 294)
> > > [2]: Enhancing Uncertainty-Based Hallucination Detection with Stronger Focus (2023)
> > > [3]: Shifting Attention to Relevance: Towards the Predictive Uncertainty Quantification of Free-Form Large Language Models (2024)*
> > >
> > > We thank the reviewer for suggesting these baselines; we were not previously aware of them, and we agree that they are relevant.
> > >
> > > During the rebuttal period, we evaluated our probes on the HalluEntity benchmark (using Llama-3.1-8B-Instruct) to compare with these methods (by directly comparing with numbers reported in [1]).
> > >
> > > However, we do want to note that evaluation on HalluEntity is quite different from evaluation on our dataset, as HalluEntity treats all words as entities to evaluate (e.g., in the sentence “Lanny Flaherty is an American actor”, the substrings [“Lanny Flaherty”, “is”, “an”, “American”, “actor”] would be treated as entities to be scored). The evaluation procedure that we use in our manuscript, on the other hand, only evaluates entities (e.g., we would only check [“Lanny Flaherty”, “American actor”], and do not score non-entity tokens such as “is” or “an”).
> > >
> > > Evaluating on HalluEntity, we find that our LoRA probe outperforms SAR, while underperforming Focus (stars indicate that numbers are taken directly from Table 2 of the HalluEntity paper):
> > > | Method | AUROC | F1_opt |
> > > | :--- | :--- | :--- |
> > > | Likelihood* | 0.584 | 0.296 |
> > > | Entropy* | 0.592 | 0.286 |
> > > | CCP* | 0.564 | 0.261 |
> > > | SAR* | 0.684 | 0.351 |
> > > | Focus* | 0.783 | 0.483 |
> > > | Our Llama 3.1 8B probe (Linear) | 0.653 | 0.330 |
> > > | Our Llama 3.1 8B probe (LoRA) | 0.721 | 0.403 |
> > >
> > > We also want to caveat these comparisons with the fact that we tried to replicate the baseline numbers for Likelihood and Entropy using our own setup, and were not able to match the results reported in Table 2 of the HalluEntity paper. Thus, we are not confident in these comparisons, and have not included them in the revised version of our paper yet. We are reaching out to the authors of HalluEntity in order to pin down discrepancies between our evaluation setups.
> > >
> > > We would additionally like to note that SAR and Focus are significantly more complex than our probing methods:
> > > - SAR is post-hoc, as it requires the complete sentence context to compute embedding shifts for each token in the sentence. It also depends on an external system to encode the text (e.g., RoBERTa).
> > > - Focus requires access to attention scores; this prevents its immediate compatibility with high-throughput inference stacks, as in high-throughput serving engines (e.g., vLLM), attention matrices are optimized away via FlashAttention/PagedAttention and are not materialized in memory. Focus also requires an external system for entity detection (e.g., SpaCy), and requires computing token IDFs from an external dataset (e.g., RedPajama).
> > >
> > > In contrast, our method is conceptually extremely simple to understand, extremely cheap to run, runs in real-time as a streaming classifier, and achieves good performance. We think that our method can likely be improved upon via more complex engineering tricks (e.g., tricks such using SpaCy, or IDF, or leveraging attention can probably squeeze out better accuracy), but we see the performance that we get from a simple methodology (arguably the simplest ML methodology - we’re simply training a logistic regression) is impressive, and a solid result for future work to build up on.

---

> > > > ### Author Response · Authors · 2025-11-23
> > > >
> > > > > *Lack of deep analysis. This paper introduced an entity-level dataset, while the experiments only focus on performance comparison without a deeper analysis on how the performance varies across different entities or what entities are easier to be hallucinated by LLMs....*
> > > >
> > > > Thank you for the suggestion; we agree that analyzing the failure-cases of our probes is important, and could help guide future improvements over our current methods.
> > > >
> > > > During the rebuttal period, we have analyzed failure cases of our probes. Overall, we didn’t qualitatively notice clear patterns in failure cases of the probes. We also performed quantitative analysis: we split entities by category, and computed per-category probe performance, and did not see significant discrepancies between categories. This suggests that the failure cases of our probes are fairly uniform or random, at least between entity categories. We have added a discussion of this analysis to the appendix (see Appendix H.3).
> > > >
> > > > Per-category performance on Longfact++ test set:
> > > >
> > > > |NER tag |Llama3 1 8b linear probe (AUC)|Llama3 1 8b LoRA probe (AUC)|
> > > > |--------|------------------|-------------------------------|
> > > > |CARDINAL|0.7988            |0.8501                         |
> > > > |DATE    |0.8507            |0.8751                         |
> > > > |GPE     |0.7155            |0.7689                         |
> > > > |MONEY   |0.7935            |0.8697                         |
> > > > |NORP    |0.7596            |0.7951                         |
> > > > |ORG     |0.7496            |0.7944                         |
> > > > |PERCENT |0.7231            |0.8837                         |
> > > > |PERSON  |0.7900            |0.8355                         |
> > > > |ART     |0.8250            |0.8790                         |
> > > >
> > > > ---
> > > >
> > > > **Addressing questions**
> > > >
> > > > > *It would be clearer to consistently use either token-level or entity-level within the whole paper.*
> > > >
> > > > We respectfully disagree, as we use token-level and entity-level to mean different things. When we say token-level, we are generally referring to our probes and probe training - the probes operate at the individual token level. When we are annotating data within our pipeline, we extract entities and label hallucinations at the entity-level; but these entity-level labels must ultimately be converted to token-level labels in order to train our probes.
> > > >
> > > > We hope that Figure 1 makes this difference clear: the probe scores are per-token (as demonstrated by the various token-level probe scores, even within a single entity); the ground-truth labels are per-entity (e.g., composed of multiple tokens; underlined in red or green).
> > > >
> > > > That being said, upon re-reading our manuscript, we identified some imprecise usage of “token-level” vs “entity-level”. We have revised the manuscript to be more precise in our language, and we appreciate the reviewer pointing out this issue.
> > > >
> > > > > *As shown in [4], there may be sentence- and relation-level hallucinations. How does the proposed pipeline handle these types of hallucination? [4]: Fine-grained Hallucination Detection and Editing for Language Models (2024)*
> > > >
> > > > In our annotation pipeline, we do not specify that Claude label sentence- and relation-level hallucinations. We instruct Claude to extract and annotate entities within the text.
> > > >
> > > > However, given that our instructions are fairly flexible, we find that Claude sometimes labels broader spans when there’s a non-factuality and it is not clearly linked to an entity. For example, in Figure 6, the whole phrase “lithium, valproate, or carbamazepine as first-line treatment for postpartum mania” is labeled by Claude as an entity.
> > > >
> > > > Additionally, when annotating completions, Claude has access to the entire completion - the completion is not split into atomic facts - and thus Claude can assess whether certain statements are true in light of the previous context of the completion.
> > > >
> > > > We would also like to note that, despite our annotation pipeline and training data being focused on entity-level hallucinations, we observe that, surprisingly, our probes generalize to other domains like detecting incorrect mathematical reasoning.

---

> > > > > ### Author Response · Authors · 2025-11-23
> > > > >
> > > > > We sincerely thank the reviewer for an extremely thoughtful and helpful review. We think many of the points are very good, and feel that the revised manuscript has been significantly improved as a result of this reviewer's feedback.
> > > > >
> > > > > We kindly ask the reviewer to consider adjusting their rating based on our revisions and clarifications.

---

> ### Comment · Reviewer_QrDD · 2025-11-26
>
> I thank the authors for their detailed responses, which addressed many of my concerns. However, I still have concerns about the **definition of entity** and **clarity**.
>
> - **Definition of entity**: The authors claimed that the relatively flexible definition of entity prevents the annotation process from overfitting. However, I think a clearer definition (beyond example types) is still necessary to curate a high-quality dataset, especially, as shown in Figure 6, when the LLM annotator sometimes considers a long span (e.g., a sentence) as an entity. Since the motivation of entity- (or token-) level hallucination detection in long-form generation is to precisely pinpoint the hallucinated content rather than marking the whole generation as hallucination, labeling a long span as hallucination may make the entity-level dataset less meaningful, as the long span may contain many tokens that are indeed not hallucinated. This may also reduce the reliability of the annotation and the authors' validation process. Although the authors showed that the LLM annotator has a high recall rate for identifying hallucinated entities, this could be easily achieved by extracting long spans, as each span is more likely to contain hallucinated tokens. Apart from the definition, I would also like to see the statistics of the entity length to resolve this concern.
> - **Clarity**: The usage of "token-level" and "entity-level" is clearer now. However, I think it would be much clearer if the author could provide a problem statement, which contains the mathematical introduction of entity-level and token-level labels, and how the performance is evaluated.
>
> I would also like to kindly ask the authors to highlight their revision with a color in the manuscript so that it is easier to check.

---

> > ### Author Response · Authors · 2025-12-01
> >
> > We sincerely thank the reviewer for engaging in the rebuttal period. We are happy to hear that many of your concerns are addressed by our initial response, and we are grateful for your continued engagement to help us improve our manuscript. We address the remaining two concerns below.
> >
> > ---
> >
> > *Definition of entity…*
> >
> > - We agree with the reviewer that the minimality and precision of entity spans is important for the validity of our approach. If the pipeline were to simply label whole sentences as individual entities, the benefit of granular detection would be lost.
> > - To address this, we have added a quantitative analysis of span lengths to Appendix C.4. The majority of extracted spans are short, with a median span length of 5 tokens and a 90th percentile of 13 tokens (see Figure 9). We additionally include a random sample of 50 entities extracted by our pipeline (Figure 10), which gives a concrete sense of the annotations produced. These statistics align with our operational definition, which explicitly instructs the annotator to “extract the minimal text span (just the entity itself, not surrounding context)” (Appendix C.5).
> > - That said, the distribution does reveal a long tail of longer spans. The long span noted in Figure 6 represents a case where no specific entity is hallucinated, but the relationship or claim surrounding it is. In that example, the drugs (entities) are real, but the claim that they are “first-line treatment for postpartum mania” is false. The annotation pipeline labeled the full claim because there was no single atomic entity that contained the falsehood. This illustrates both the limitation of the entity-based approach (not all hallucinations can be pinpointed to a single fabricated entity) and the flexibility of our pipeline in catching broader errors when they arise.
> > - We acknowledge the reviewer’s valid concern that high recall could theoretically stem from over-selecting long spans. However, our data precludes this explanation: we achieve 80.6% recall on injected hallucinated entities while maintaining a median span length of just 5 tokens. This confirms that our pipeline achieves high recall through precise localization, rather than by annotating long spans to capture broad contexts.
> > - We have revised our Limitations section (Section 6) to be explicit about this trade-off, and the new Appendix C.4 provides the quantitative analysis and discussion.
> >
> > *Clarity…*
> >
> > - We agree that a formal problem statement would significantly improve the clarity of the paper. We have revised Section 3.2 to include a mathematical formulation that explicitly defines entity spans ($s \in \mathcal{S}$) and labels ($y_s \in {0,1}$), token-level predictions ($p_i$), and the aggregation function used to bridge between the two levels.
> > To further reinforce this distinction, we have also renamed the methodology subsections:
> > - Section 3.1: Dataset construction and **entity-level annotation**
> > - Section 3.2: Problem formulation and **token-level probe training**
> >
> > *… highlight their revision with a color…*
> >
> > - We have updated the revised manuscript so that all revisions are now **indicated in blue**.
> >
> > ---
> >
> > We again thank the reviewer for their continued engagement. We genuinely feel that our manuscript has been significantly improved thanks to their thoughtful questions and suggestions.

---

### Official Review · Reviewer_arys · 2025-10-16

**Soundness:** 2
**Presentation:** 3
**Contribution:** 3
**Rating:** 4
**Confidence:** 5

**Summary:**

The paper proposes a streaming, token‑level detector for hallucinated entities (names, dates, citations) in long‑form LLM outputs by training lightweight linear/LoRA probes on hidden states; entity labels are obtained via an LLM‑with‑web‑search pipeline, then mapped to tokens and optimized with a combined tokenwise + span‑max loss. The approach achieves strong long‑form performance and generalizes to short‑form QA and even math reasoning without explicit entities. It further shows cross‑model transfer and explores selective answering by abstaining when probe scores cross a threshold, improving conditional accuracy at the cost of attempt rate. The authors discuss KL‑regularized LoRA to retain model behavior while keeping high detection AUC and commit to release code/data upon publication

**Strengths:**

Practical, low‑latency design: token‑level probes run in the forward pass and support real‑time detection during generation; entity focus preserves token alignment for streaming.

Consistent long‑form gains: On LongFact / LongFact++ / HealthBench, linear probes ≥0.85 AUC; LoRA ≈0.89–0.93, beating perplexity/entropy/semantic‑entropy baselines.

Robust generalization: Trained on long‑form, probes transfer to short‑form (TriviaQA LoRA AUC ≈0.97–0.98) and to MATH reasoning (AUC ≈0.88), suggesting the signal captures more than entity fabrication.

Cross‑model utility: Probes trained on one model detect hallucinations in others with small degradation; larger‑model probes supervise smaller models effectively.

Deployment‑oriented analysis: KL‑regularized LoRA offers a tunable trade‑off between detection AUC and output distribution shift; authors quantify KL, win‑rate, and MMLU impacts.

Selective answering demo: Monitoring probe scores to abstain boosts conditional accuracy (e.g., Llama‑3.3‑70B from 27.9% → 50.4%) illustrating actionable safety benefits.

**Weaknesses:**

Label noise & ceiling: Annotation pipeline achieves ~80.6% recall on synthetic errors with ~15.8% FPR; this noise constrains both training effectiveness and evaluation confidence.

Recall at practical FPR still modest: On long‑form, R@0.1 ≈0.70 (often 0.65–0.78), which may be insufficient for high‑stakes deployments where misses are costly.

Entity‑centric scope: Focus on entity fabrication misses broader reasoning/relational hallucinations; evidence of MATH transfer is promising but indirect.

Annotation cost & dependence: Data creation relies on a frontier LLM with web search; while runtime is cheap, upfront labeling is expensive and may depend on proprietary tools whose behavior can drift.

Behavioral side‑effects: Unregularized LoRA can materially shift output distributions (lower win‑rates/MMLU); even with KL regularization, careful calibration is required per deployment.

Utility trade‑off in abstention: Selective answering significantly reduces attempt rate (e.g., 76.1% → 19.1% for Llama‑3.3‑70B at t=0.5), which may limit usefulness in production.

**Questions:**

Annotation efficiency: Can you quantify the cost/time per annotated long‑form sample and explore semi‑supervised or active‑learning loops to reduce reliance on search‑augmented LLM judges?

Beyond entities: How would you extend the streaming detector to claim‑ or relation‑level hallucinations (reasoning chains), while maintaining token alignment and low latency?

Thresholding & calibration: How should production systems set and adapt thresholds (per domain/user risk) to balance R@FPR trade‑offs, especially under distribution shift?

Harm‑aware detection: Can the probe be trained to weight hallucinations by potential harm, prioritizing medically/legal‑impactful entities over inconsequential ones?

Robustness of cross‑model transfer: Do probes trained on larger, more up‑to‑date models degrade when evaluating stronger future models where “supported” knowledge expands? Any plans for continual calibration?

Human evaluation: Beyond AUC/R@FPR, do you have human‑in‑the‑loop studies showing reductions in user‑visible factual errors in realistic workflows (multi‑turn, mixed modality)?

---

> ### Author Response · Authors · 2025-11-23
>
> We thank reviewer arys for their review. We appreciate their acknowledgement that our work provides a practical, low-latency framework for hallucination detection, and that we study various aspects of the system comprehensively.
>
> Overall, we broadly agree with the weaknesses that you have pointed out, and already explicitly highlight all of them as limitations in our manuscript. We see this work as an initial step towards real-time streaming hallucination detection, and not yet a production-ready solution. As such, we tried to be forthcoming and honest about the limitations of our work in our manuscript. In fact, while our original submission had discussed our limitations in the appendix, we have now brought it into the main body for additional clarity.
>
> ---
>
> **Addressing weaknesses**
>
> > *Label noise & ceiling; Annotation cost & dependence*
>
> We agree that the cost and accuracy of the annotation pipeline is a limitation of our work. We acknowledge these explicitly in our manuscript.
>
> To address the limitation of cost, we (1) study cross-model generalization, and show that annotated datasets from one model’s generations can be used to effectively train another model as a hallucination probe; and (2) release our dataset open-source, to make the methodology cheap and more accessible for future work.
>
> We also did diligence to study the quality of our annotation pipeline, and discuss the results in Appendix G. To study this, we (1) conducted a study to check agreement between human annotators and the annotation pipeline; (2) checked precision and recall for synthetically generated hallucinations, for which we do have ground-truth; and (3) checked consistency between different annotation models.
>
> Overall the performance of our pipeline was impressive, although obviously not perfect, and we try to honestly report this limitation. For the purposes of training linear probes, we find the dataset quality to be sufficient to achieve high performance on the task of hallucination detection, and show that it outperforms various baselines from the literature.
>
> > *Recall at practical FPR still modest; Utility trade‑off in abstention*
>
> We agree that our method is still imperfect, and does not completely solve the problem of hallucination detection. We acknowledge this in our manuscript explicitly.
>
> However, we do think our work is a step in the direction of progress, as our methodology achieves good performance (out-performs prominent baselines such as Semantic Entropy) while also being extremely cheap and running in real time. We hope that future work can build off of this work, and improve reliability of detection.
>
> > *Entity‑centric scope*
>
> We acknowledge this limitation explicitly in our manuscript. We do note that, although we train our probes only to detect entity-level hallucinations, we find that they generalize to out-of-domain distributions such as mathematical reasoning.
>
> > *Behavioral side‑effects*
>
> The linear probes (without LoRA modifications) achieve good performance and have no side-effects on behavior. The LoRA probes boost classification accuracy, but exhibit a tradeoff in modifying the model’s behavior. This tradeoff is discussed and studied extensively in Appendix J.

---

> > ### Author Response · Authors · 2025-11-23
> >
> > **Addressing questions**
> >
> > > *Annotation efficiency: Can you quantify the cost/time per annotated long‑form sample and explore semi‑supervised or active‑learning loops to reduce reliance on search‑augmented LLM judges?*
> >
> > Thank you for this question - we’ve updated the manuscript (Appendix L) to include the approximate cost of our annotation pipeline.
> >
> > We’re not sure what you mean by “explore semi-supervised or active-learning loops” - could you explain what you have in mind here?
> >
> > > *Beyond entities: How would you extend the streaming detector to claim‑ or relation‑level hallucinations (reasoning chains), while maintaining token alignment and low latency?*
> >
> > In our discussion section, we explicitly mention this challenge as an exciting avenue for future work
> >
> > While in this work we focus on entity-level hallucination detection, we actually did find that our probes generalized to broader classes of hallucinations, such as detecting faulty reasoning in the math domain. This suggests that training to detect entity-level hallucinations can, at least to some extent, transfer to other types of hallucination. While this is not the main focus or claim of our manuscript, we are excited for future work to explore this more.
> >
> > > *Thresholding & calibration: How should production systems set and adapt thresholds (per domain/user risk) to balance R@FPR trade‑offs, especially under distribution shift?*
> >
> > This is a fair question. In real-world applications, it would be up to the user to set their thresholds. Different domains have different risk profiles. For example, in the medical or legal domain, incorrect information is extremely dangerous, and so users could tweak their detectors to have high recall, even if it means having a high false-positive rate. In other domains, such as chatbot settings, thresholds might be set to be more permissive, since the stakes are lower.
> >
> > > *Harm‑aware detection: Can the probe be trained to weight hallucinations by potential harm, prioritizing medically/legal‑impactful entities over inconsequential ones?*
> >
> > This is an interesting question for future work. We think a plausible solution here would be to have another classifier or system that judges the risk (e.g., it can distinguish between high risk and low risk contexts), and then based on this additional risk-assessment system, dynamically adjust the threshold of the hallucination detection probes. This would allow for more sensitive hallucination detection in high-risk scenarios, without necessitating retraining of probes.
> >
> > > *Robustness of cross‑model transfer: Do probes trained on larger, more up‑to‑date models degrade when evaluating stronger future models where “supported” knowledge expands? Any plans for continual calibration?*
> >
> > The models we tested for cross-model have roughly similar knowledge cutoff dates. As the field moves forward, and new models are released with more recent knowledge cutoffs, the discrepancy between knowledge cutoffs will become an important factor for cross-model transfer of hallucination detection.
> >
> > > *Human evaluation: Beyond AUC/R@FPR, do you have human‑in‑the‑loop studies showing reductions in user‑visible factual errors in realistic workflows (multi‑turn, mixed modality)?*
> >
> > This is a great idea; but we do not have this in scope for the current manuscript. We are excited for future work to take hallucination detection methods, and turn them into solutions that reduce rates of hallucination (for example, by resampling until a hallucination detector classifies the content to be non-hallucinated).
> >
> > ---
> >
> > We again thank the reviewer for their review. We hope that our responses have clarified our work, and resolved the concerns. We kindly ask that the reviewer revise their rating in light of our thorough response.
> >
> > If you have any further suggestions that would lead to an improved rating, we would be grateful for that feedback.

---

### Official Review · Reviewer_tiwT · 2025-10-27

**Soundness:** 2
**Presentation:** 2
**Contribution:** 2
**Rating:** 2
**Confidence:** 4

**Summary:**

The paper proposes a token-level, real-time hallucination detection framework for large language models (LLMs). By framing hallucination detection as a sequence labeling task, the authors train lightweight linear and LoRA probes to identify fabricated entities in long-form generation. A new dataset, LongFact++, is introduced using web-augmented annotation with Claude-4 Sonnet. Experiments across five model families show consistent AUC gains over baselines like semantic entropy and token-level perplexity. The method generalizes to different models and even to out-of-domain reasoning tasks such as MATH. The authors also demonstrate a real-time selective-answering setup that abstains when hallucination risk is high.

**Strengths:**

- It is quite convincing that framing the hallucination detection as a token-level task for streaming detection is original and practically motivated.
- empirical results seem effective; The proposed probes outperform semantic entropy and uncertainty-based baselines with clear quantitative evidence.

**Weaknesses:**

- **Questionable conference fit**: The paper is more application-oriented and empirical than methodologically innovative, making it feel more aligned with NLP venues (ACL, EMNLP, NAACL) rather than ICLR, which prioritizes fundamental learning representations or optimization insights.
- **limited novelty in the methodology**: The core technique—linear or LoRA probes trained on hidden states—is not conceptually new; the contribution lies mainly in data engineering and evaluation. Thus, the review was concerned that constructing an efficient training dataset alone truly gives sufficient information to handle hallucination detection.
- **Annotation pipeline**: The web-based LLM annotation method is expensive, noisy, and partially unverifiable. Reported human agreement (84%) and false positive rates (~15%) suggest quality issues that may affect probe robustness.
- **Evaluation depth**: The study lacks qualitative analysis of failure cases and does not compare against strong retrieval-augmented verification systems under similar latency constraints.
- **Overstated generalization**: While probes show some transfer to reasoning tasks, it’s unclear whether this reflects genuine factuality awareness or spurious correlations in hidden representations.

**Questions:**

Good experimental rigor and motivation, but lacks theoretical depth and venue alignment.
The work would be a stronger fit for an NLP-focused conference emphasizing factuality and evaluation rather than ICLR’s learning-theoretic orientation.

---

> ### Author Response · Authors · 2025-11-23
>
> We thank reviewer tiwT for their review.
>
> We appreciate the acknowledgement that our framing of long-form hallucination detection is “original and practically motivated”, and also that our “empirical results seem effective”.
>
> ---
>
> **Addressing weaknesses**
>
> > *Questionable conference fit…*
>
> We appreciate the reviewer’s suggestion that our work might be a better fit for other conferences. However, we respectfully disagree that the topic of our paper is not a fit for ICLR; on the contrary, we think that our work would be of interest to many researchers within the ICLR community.
>
> ICLR’s [official call for papers](https://iclr.cc/Conferences/2026/CallForPapers) is very broad, as they include a “broad range of subject areas including … uncertainty quantification and issues regarding large-scale learning…, as well as applications in … language, … and others.”
>
> Additionally, the study of hallucinations in LLMs was a prominent topic at last year’s ICLR:
> - There was an entire ICLR workshop dedicated to hallucination research: [Quantify Uncertainty and Hallucination in Foundation Models: The Next Frontier in Reliable AI](https://iclr.cc/virtual/2025/workshop/23965).
> - There was an *oral* awarded at last year’s ICLR for research on hallucination: [Do I Know This Entity? Knowledge Awareness and Hallucinations in Language Models](https://openreview.net/forum?id=WCRQFlji2q) (Ferrando & Obeso et al., 2025); a *spotlight*: [Reducing Hallucinations in Large Vision-Language Models via Latent Space Steering](https://openreview.net/forum?id=LBl7Hez0fF) (Liu et al., 2025); and many posters, including closely related work on probing for hallucinations: [LLMs Know More Than They Show: On the Intrinsic Representation of LLM Hallucinations](https://openreview.net/forum?id=KRnsX5Em3W) (Orgad et al., 2025).
>
> Further, our approach to hallucination detection leverages the *internal representations* of language models; our methodology is not simply a black-box pipeline, but rather builds off of work showing that models have some internal representations of confidence or true knowledge. Our work shows that these representations are so strong that they can potentially actually be *useful* for solving a real-world problem: hallucination detection - a problem that people really care about.
>
> > *Limited novelty in the methodology: The core technique—linear or LoRA probes trained on hidden states—is not conceptually new…*
>
> We agree that the use of linear probes is not novel; indeed it is a well-established technique in the literature. Rather, we see the main contribution of our paper as taking a real-world problem (long-form hallucination detection), formulating it as a token-level classification task, introducing a token-level annotation pipeline for dataset creation, and then rigorously evaluating the resulting probes. We think that this type of applied work on a real-world problem is a valuable contribution, even if the methodologies we use are simple and well-established.
>
> If there is any wording that you think we could modify in our manuscript to make this point clear (namely, that the use of linear probes is not novel), we are happy to revise it. We felt like the first paragraph of our “Related work” section frames this properly, as we discuss prior works that use linear probes; but we are open to more specific suggestions of how it could be made clearer.
>
> Separately, could you please clarify what you meant by this sentence from your review? We are having trouble parsing its meaning, and would like to understand better what you are saying: *“Thus, the review was concerned that constructing an efficient training dataset alone truly gives sufficient information to handle hallucination detection.”*

---

> > ### Author Response · Authors · 2025-11-23
> >
> > > *Annotation pipeline… is expensive, noisy, and partially unverifiable…*
> >
> > We agree that the cost and accuracy of the annotation pipeline is a limitation of our work. We acknowledge these explicitly in our manuscript.
> >
> > To address the limitation of cost, we (1) study cross-model generalization, and show that annotated datasets from one model’s generations can be used to effectively train another model as a hallucination probe; and (2) release our dataset open-source, to make the methodology cheap and more accessible for future work.
> >
> > We also did diligence to study the quality of our annotation pipeline, and discuss the results in Appendix G. To study this, we (1) conducted a study to check agreement between human annotators and the annotation pipeline; (2) checked precision and recall for synthetically generated hallucinations, for which we do have ground-truth; and (3) checked consistency between different annotation models.
> >
> > Overall the performance of our pipeline was impressive, although obviously not perfect, and we try to honestly report this limitation. For the purposes of training linear probes, we find the dataset quality to be sufficient to achieve high performance on the task of hallucination detection, and show that it outperforms various baselines from the literature.
> >
> > > *Evaluation depth: The study lacks qualitative analysis of failure cases and does not compare against strong retrieval-augmented verification systems under similar latency constraints.*
> >
> > Could you please elaborate on which “strong retrieval-augmented verification systems” have “similar latency constraints” to our probing method? One of the main points of our paper is that retrieval-based verification systems inevitably incur high cost and latency. Our probe-based method, on the other hand, is extremely cheap (it can be computed with negligible cost, as a single linear projection from a hidden state), and enables real-time monitoring (with no latency penalty). Thus, we are unaware of any retrieval-based methods with “similar latency constraints”.
> >
> > During the rebuttal period, we have analyzed failure cases of our probes. Overall, we didn’t qualitatively notice patterns in failure cases of the probes. We also performed quantitative analysis: we split entities by category, and computed per-category probe performance, and did not see significant discrepancies between categories. This suggests that the failure cases of our probes are fairly uniform or random. We have added a discussion of these experiments to the appendix (see Appendix H.3).
> >
> > Per-category performance on Longfact++ test set:
> >
> > |NER tag |Llama3 1 8b linear probe (AUC)|Llama3 1 8b LoRA probe (AUC)|
> > |--------|------------------|-------------------------------|
> > |CARDINAL|0.7988            |0.8501                         |
> > |DATE    |0.8507            |0.8751                         |
> > |GPE     |0.7155            |0.7689                         |
> > |MONEY   |0.7935            |0.8697                         |
> > |NORP    |0.7596            |0.7951                         |
> > |ORG     |0.7496            |0.7944                         |
> > |PERCENT |0.7231            |0.8837                         |
> > |PERSON  |0.7900            |0.8355                         |
> > |ART     |0.8250            |0.8790                         |
> >
> >
> > > *Overstated generalization: While probes show some transfer to reasoning tasks, it’s unclear whether this reflects genuine factuality awareness or spurious correlations in hidden representations*
> >
> > Could you please elaborate on what you mean by “it’s unclear whether this reflects genuine factuality awareness or spurious correlations in hidden representations”?
> >
> > We’d note that the probes were trained with *no math data*; and yet when we evaluate on the math domain, we find that the probes generalize well to detecting incorrect math statements (see Figure 7 in Appendix C.2 for a concrete example). We take this as some evidence that the probes are picking up on some true underlying signal of factuality; we think if the probe were to pick up on spurious correlations in the training data, then it would be less likely to generalize to out-of-distribution domains such as math.
> >
> > Overall, we do acknowledge the limitations of our entity-based detection method, and acknowledge this in our limitations section. However, we thought it was important to present the surprising generalization results on math, given that prior works on probe-based hallucination detection report poor generalization on the math domain (e.g., Orgad et al., 2025).
> >
> > ---
> >
> > We again thank the reviewer for their review. We hope that our responses have clarified our work, and resolved the concerns. We kindly ask that the reviewer revise their rating in light of our thorough response.
> >
> > If you have any further suggestions that would lead to an improved rating, we would be grateful for that feedback.

---

### Official Review · Reviewer_H1ec · 2025-11-01

**Soundness:** 3
**Presentation:** 2
**Contribution:** 2
**Rating:** 6
**Confidence:** 4

**Summary:**

The goal of the paper is to detect hallucinations in long form generations. The motivation of the paper is that previous methods are mostly limited to short form generations and require manual verification. The paper scales to larger generations by considering entity level hallucinations, that is, hallucinations in names, citations etc, rather than focusing on claim level hallucinations. This focus allows the paper to map the hallucination detection task to individual tokens. The key idea is to first obtain a training dataset by performing a web search and annotate the model’s responses for correctness. Then paper then trains linear probes in a manner similar to prior works like Azaria and Mitchell and Orgad et al. The probes can also be accompanied by LoRA adapters. Interestingly, the paper shows that the method can be generalized across domains to math problems as well.

**Strengths:**

1. Extension of hallucination detection to long generations is an important problem. Much of prior work indeed seems to focus on short statements with atomic claims.
2. The experimental analysis is well-thought though. The ablations in Section 5, e.g., checking cross model and cross task transfer are quite interesting.
3. The idea of early abstention, that is, halting generating when the first hallucination is detected has potentially large practical benefits.

**Weaknesses:**

1. Currently, the paper lacks strong baselines. The paper correctly points out that prior work on long form hallucination detection has to rely on external fact checking for individual claims. However, the paper itself breaks the problem down into checking hallucinations in entities and hence checks for individual hallucinations. Potentially, one could break the whole generation down into sentences and run it through hidden-state based classifiers like those trained by Orgad et al. It would be great to compare with such baselines and compare the performance and the cost.
2. The paper seems to oversell the contribution a little bit. True that by relying on entity level hallucinations, the paper can perform detection in a streaming manner. However, it just means that the paper solves a very different problem, that is, entity hallucination as opposed to claim hallucination. For instance, some claims do not have to involve entities, e.g., logical and mathematical claims. So it would be nice to tone down the statements a little bit or provide justification of why the paper thinks it is solving the same problem.
3. The dataset contribution in extending LongFact to LongFact++ (Appendix D) seems relatively small. It is also not very clear what we obtained by gathering the new dataset. True that we get more topics. But does it lead to concrete benefits, like enhancing the cross-domain or cross-model generalization? A comparison between LongFact and LongFact++ would be very helpful in framing the dataset contribution.
4. The writing could be improved at some places. For instance, Appendix G addresses several problems but the mention of it in the main paper (line 182) is quite hurried. I would suggest bringing more of the text from Appendix D and G into the main paper so that the readers have the necessary context on what problems are being solved.

**Questions:**

1. Will the new dataset and annotations be made publicly available?

---

> ### Author Response · Authors · 2025-11-23
>
> We thank reviewer H1ec for their thoughtful review and feedback.
>
> We appreciate the acknowledgement that long-form hallucination detection is an important problem that has been relatively neglected, and that our experiments are interesting and well thought through.
>
> ---
>
> **Replying to weaknesses**
>
> > *Currently, the paper lacks strong baselines.…*
>
> Our current manuscript includes various baselines, ranging from simple ones to more complex ones. In particular, we thought it was important to compare with semantic entropy, which, as far as we know, is thought of as the current state-of-the-art method for uncertainty detection. While semantic entropy is not directly designed for entity-level detection, we took care to adapt it to the entity-level detection task (see details in Appendix G.2).
>
> For our main results (Figure 2), we include 3 baselines:
> - Token-level entropy - computing the entropy of a given token, and using that as the token’s “score”
> - Token-level perplexity - computing the perplexity of a given token, and using that as the token’s “score”
> - Semantic entropy - for each entity, we take the completion up to right before the entity, and generate 10 completions, cluster the completions by semantic similarity, and then compute the entropy over semantic clusters.
>
> Additionally, in Appendix G.3, we compare against a prompting baseline, where we ask the model whether a particular sentence contains a hallucination. For this evaluation, we check at the sentence-level (e.g., a sentence is considered hallucinated if all entities in it are hallucinated; non-hallucinated if all entities are non-hallucinated; otherwise the sentence is not evaluated on). We tried hard to steel-man this baseline, iterating on prompting setups, and we even compared to a baseline where the model is finetuned on this particular task (the task of answering the question of “does this sentence contain a hallucination?” with the correct answer).
>
> > *The paper seems to oversell the contribution a little bit.…*
>
> Thank you for this feedback. We tried to be careful not to oversell our contribution, and tried to emphasize that our work focuses on entity-level hallucinations. We titled the paper “... Detection of Hallucinated Entities…” for this reason, and also take care to emphasize this scope in our abstract, introduction, and throughout the paper. Additionally, we discuss this limitation explicitly in our Limitations section (this section was previously in our appendix, but we have recently moved it into the main body, in order to make our limitations more visible).
>
> Are there specific statements in our manuscript that you could point to that seem like overclaims? We would be grateful if you could point them out so that we can correct them! We care a lot about epistemic honesty and integrity, and would like to correct any overclaims that we may have overlooked.
>
> > *The dataset contribution in extending LongFact to LongFact++ seems relatively small.…*
>
> We see the creation of LongFact++ (our new set of prompts / questions) as only a minor contribution of our paper (e.g., it is not mentioned in the abstract). With our training-based approach to hallucination detection, we wanted a larger and more diverse dataset to support larger scale training. We see the utility of LongFact++ as mainly providing us with more data. LongFact contains only ~2k prompts, while our augmented dataset contains ~20k prompts.
>
> Figure 3 in the manuscript displays AUC as a function of training samples, and gives a sense of probe improvement as a function of dataset size.
>
> We do think it is an interesting question to ask whether topic diversity improves generalization separately from just the raw amount of training data; however we leave this question out of scope for our current study.
>
> > *The writing could be improved at some places.…*
>
> We appreciate this piece of feedback. We felt quite limited by the page limitations, and had to make difficult decisions about what to include in the main body and what to punt to the appendix. For example, our thorough analysis of the tradeoff between LoRA probe accuracy and model behavior was relegated to the appendix (Appendix H). We agree that Appendix F (formerly Appendix G; discussing the quality checks of our annotation pipeline) is important; however, we felt that to keep the paper focused it should be briefly summarized in the main body and have longer details (without length constraints) in the appendix. This is unfortunately a tradeoff we must make given the ICLR page limits. As for Appendix C (formerly Appendix D; detailing dataset construction), we think that the details of dataset construction are reasonable to leave to the appendix for readers interested in details, as we don’t think it’s critical to understand the main points of our manuscript.

---

> > ### Author Response · Authors · 2025-11-23
> >
> > **Replying to questions**
> >
> > > *Will the new dataset and annotations be made publicly available?*
> >
> > Yes, absolutely! All datasets and annotations will be made publicly available on HuggingFace after publication.
> >
> > ---
> >
> > We again thank the reviewer for their thoughtful review. We would like to make this paper as clear as possible, and would value your feedback on what additional improvements would clarify the manuscript further. If you have any additional suggestions that would lead to an improved rating, we would be grateful for that feedback.

---

### Author Response · Authors · 2025-12-01
**Summary of reviews and rebuttals for AC**

We thank all reviewers for their engagement. Given that reviewer scores cannot be updated, we provide this summary to help the AC assess how concerns were addressed.

---

**What the paper contributes.**

We present a method for real-time, streaming detection of hallucinated entities in long-form LLM generations. Prior work on hallucination detection has either focused on short-form settings (e.g., single-answer QA) or required expensive post-hoc verification pipelines that are impractical for real-time use. Our approach frames hallucination detection as a token-level classification problem, enabling lightweight linear probes that run during generation with negligible overhead. We contribute: (1) an automated annotation pipeline using web-search-augmented LLMs, (2) a large-scale dataset of 25,000+ annotated completions (to be open-sourced), and (3) extensive experiments showing our probes outperform uncertainty-based baselines, including more expensive methods such as semantic entropy (e.g., 0.90 vs 0.71 AUC on Llama-3.3-70B).

---

**Reviewer-acknowledged strengths.**

- Long-form hallucination detection is an important practical problem
  - “Extension of hallucination detection to long generations is an important problem. Much of prior work indeed seems to focus on short statements with atomic claims.” (H1ec)
  - “The idea of early abstention… has potentially large practical benefits.” (H1ec)
  - “framing the hallucination detection as a token-level task for streaming detection is original and practically motivated.” (tiwT)
- Experiments and ablations are extensive and thoughtful
  - “The experimental analysis is well-thought through” (H1ec)
  -  “extensive experiments and analysis in different settings” (QrDD)
- Open-source dataset is valuable for future research
  - “dataset is valuable and can contribute to future research” (QrDD)
-  Clear presentation
  - “the writing is clear and easy to follow” (QrDD)

---

> ### Author Response · Authors · 2025-12-01
> **Summary of reviews and rebuttals for AC (continued)**
>
> **Main concerns and how we addressed them.**
>
> We would like to highlight reviewer QrDD- this reviewer is clearly an expert in the field and provided thoughtful feedback across two rounds of discussion that significantly improved our manuscript. Before the score freeze, QrDD noted that our initial rebuttal "addressed many of [their] concerns," and then raised two follow-up points (span length statistics and a formal problem statement), both of which we have since addressed.
>
> We summarize the main concerns and our responses below; all manuscript changes are indicated in blue.
>
> *Related work (QrDD)*: Reviewer QrDD correctly noted that we failed to cite HalluEntity (Yeh et al., 2025), which also frames hallucination detection at the entity level. We thank the reviewer for pointing out this oversight, as we were not previously aware of this work and agree that it is highly relevant. We have added citations to HalluEntity in our introduction and added an entire paragraph to our Related Work section discussing their contributions and how our work differs (larger scale, simplified pipeline, probe-based methodology).
>
> *Entity definition (QrDD)*: Reviewer QrDD raised valid concerns about the flexibility of our definition of “entity”, and how this manifests in the entity extraction step of the automated annotation pipeline. We addressed this comprehensively:
> - Revised Limitations section (Section 6) to honestly discuss the trade-off: our flexible definition allows the automated annotation pipeline to catch relational errors that a rigid definition of “entity” would miss, but this also means some labeled spans are phrases rather than atomic entities. We view this as both a feature and a limitation.
> - Performed extensive analysis on the entities that were extracted from our annotation pipeline
>   - Added NER-tag breakdown of entity categories (Appendix C.3)
>   - Added quantitative span length analysis (Appendix C.4): median 5 tokens, 90th percentile 13 tokens, confirming the pipeline produces primarily short, atomic annotations
>   - Added a random sample of 50 extracted entities (Figure 10) for transparency
> - Clarified the entity-level vs token-level distinction formally in Section 3.2, for improved clarity
>
> *Baselines (QrDD, H1ec)*: Reviewers suggested we lacked strong baselines. We respectfully note that our experiments include multiple baselines spanning different methodological approaches:
> - Token-level entropy and perplexity (standard uncertainty metrics)
> - Semantic entropy ([Kuhn et al., 2023](https://openreview.net/forum?id=VD-AYtP0dve)), which we carefully adapted to entity-level detection by sampling continuations from entity boundaries and clustering by semantic equivalence (details in Appendix G.2)
> - Black-box self-evaluation, where we prompt the model to judge whether its own sentences contain hallucinations; we tried hard to steelman this baseline, iterating on prompting setups and even fine-tuning the model on this specific task (Appendix G.3)
>
> *Conference fit (tiwT)*: Reviewer tiwT suggested the paper might be better suited for NLP venues. We respectfully disagree: ICLR 2025 featured an oral on hallucination detection ([Ferrando & Obeso et al., 2025](https://openreview.net/forum?id=WCRQFlji2q)), a spotlight on hallucination reduction ([Liu et al., 2025](https://openreview.net/forum?id=LBl7Hez0fF)), and a dedicated [workshop on hallucinations in foundation models](https://iclr.cc/virtual/2025/workshop/23965). Our work additionally leverages internal model representations, connecting to the interpretability literature that is well-represented at ICLR.
>
> ---
>
> **Summary.**
>
> We have made substantial revisions to the manuscript in response to reviewer feedback, all indicated in blue. To take advantage of the extra page allowed during rebuttals, we elected to move our Limitations section from the appendix into the main body (Section 6), where we provide what we believe is an honest and thorough self-assessment of the limitations of our work. We invite the AC to read this section.
> With these limitations acknowledged, we believe this work makes a meaningful contribution: a simple, cheap, real-time method for detection of hallucinated entities in long-form generation that outperforms existing approaches. We will open-source our dataset of 25,000+ annotated completions, which we hope will enable future research on this important problem. We believe this work will be of interest to many researchers in the ICLR community.

---

### Meta-Review · Area_Chair_yePi · 2026-01-08

**Summary:**

Reviewers’ concerns centered on whether the paper’s strong empirical claims are sufficiently supported given (i) **baseline completeness**—multiple reviewers felt the comparison set relies heavily on uncertainty heuristics (entropy/perplexity/semantic entropy) and lacks strong, task-aligned token/entity-level detectors or retrieval-style verifiers under matched latency constraints, weakening the “probes markedly outperform baselines” conclusion; (ii) **annotation pipeline reliability and reproducibility**—the web-search–augmented LLM labeling pipeline is costly, partially unverifiable, and demonstrably noisy (non-trivial FPR / imperfect recall), raising doubts about dataset quality, evaluation ceiling, and how robust the learned signal is; (iii) **limited methodological novelty / venue fit**—the core technique (linear/LoRA probing of hidden states) is not new, with novelty primarily in problem framing, data engineering, and empirical study, which some reviewers viewed as better aligned with NLP venues unless the methodological contribution is strengthened; and (iv) **scope clarity**—while the work targets entity-level hallucinations, reviewers worried about overgeneralizing to broader claim/reasoning hallucinations and wanted clearer definitions, analyses, and justification for transfer results.

**Reviewer Concerns:**

Addressed by the rebuttal: The authors substantially improved related-work coverage and positioning (adding and discussing HalluEntity), clarified the token-level vs entity-level setup with a more formal problem formulation, and directly responded to concerns about the entity definition by adding transparency artifacts (NER-type breakdown, random samples) and span-length statistics (showing most spans are short while acknowledging a long-tail). They also expanded and foregrounded limitations, explicitly acknowledging the entity-centric scope, cases where relational errors yield longer spans, and the cost/noise trade-offs of the annotation pipeline, and they added additional analyses (e.g., per-entity-category performance) to improve depth and clarity.

Still outstanding: The most consequential gap remains baseline completeness under matched conditions: reviewers requested stronger token/entity-level baselines (e.g., Focus/SAR-style or comparable token-level detectors) and/or stronger verification-style systems under realistic latency constraints; the rebuttal discusses these but does not convincingly integrate them into the core evaluation, leaving the “outperforming baselines” claim somewhat fragile. Second, while audits help, annotation reliability and reproducibility are still fundamental concerns because the pipeline depends on proprietary, potentially drifting web-search–augmented LLM judgments and remains imperfect (non-trivial FPR / missed extractions), limiting how definitive the conclusions are. Finally, the paper still risks scope/claim mismatch: evidence for “beyond-entity” generalization (e.g., math/reasoning transfer) is suggestive but not deeply explained or controlled, so broad claims about hallucination detection beyond entities should remain carefully bounded.

**Reviewer Scores:**

Reviewer QrDD (score 4 / marginally below): Likely up to 6. Their main actionable requests (cite/discuss HalluEntity, clearer entity vs token formulation, span-length statistics, more clarity) were directly addressed, so a move into borderline accept is plausible, though they may still want stricter entity definition and stronger in-paper baselines.

Reviewer H1ec (score 6 / marginally above): Likely stay 6. The rebuttal clarifies scope (entity-level) and argues baseline strength, but their core ask—stronger, comparable baselines beyond uncertainty measures—was only partially satisfied, so any increase would be modest.

Reviewer arys (score 4 / marginally below): Likely 4. They already viewed it as borderline; added cost quantification, clearer limitations, and extra analyses reduce uncertainty, but concerns about label noise, recall at practical FPR, and deployment trade-offs would probably prevent a jump to a confident accept.

Reviewer tiwT (score 2 / reject): Likely 2. Their objections are largely about venue fit and limited methodological novelty; rebuttal clarifications help, yet the underlying preference for more fundamental ML contribution likely remains.

---

### Decision · Program_Chairs · 2026-01-26

Reject